



# CMIP6 simulations with the compact Earth system model OSCAR v3.1

Yann Quilcaille[1*], Thomas Gasser[2], Philippe Ciais[3], Olivier Boucher[4]

[1] International Institute for Applied Systems Analysis (IIASA), 2361 Laxenburg, Austria;
[*] now at Institute for Atmospheric and Climate Science, Department of Environmental Systems Sciences, ETH Zürich,
5  Zürich, Switzerland
[2] International Institute for Applied Systems Analysis (IIASA), 2361 Laxenburg, Austria
[3] Laboratoire des Sciences du Climat et de l'Environnement, LSCE/IPSL, Université Paris-Saclay, CEA – CNRS – UVSQ,
91191 Gif-sur-Yvette, France
[4] Institut Pierre-Simon Laplace, Sorbonne Université, CNRS 75252 Paris, France

10  Correspondence to: Yann Quilcaille (yann.quilcaille@env.ethz.ch)

**Abstract.** While Earth system models (ESMs) are process-based and can be run at high resolutions, they are only limited by computational costs. Reduced complexity models, also called simple climate models or compact models, provide a much cheaper alternative, although at a loss of spatial information. Their structure relies on the sciences of the Earth system, but with a calibration against the most complex models. Therefore it remains important to evaluate and validate reduced 15 complexity models. Here, we diagnose such a model the newest version of OSCAR (v3.1) using observations and results from ESMs from the current Coupled Model Intercomparison Project 6. A total of 99 experiments are selected for simulation with OSCAR v3.1 in a probabilistic framework, reaching a total of 567,700,000 simulated years. A first highlight of this exercise that the ocean carbon cycle of the model may diverge under some parametrizations and for high-warming scenarios. The diverging runs caused by this unstability were discarded in the post-processing. Then, each physical parametrization is 20 weighted based on its performance against a set of observations, providing us with constrained results. Overall, OSCAR v3.1 shows good agreement with observations, ESMs and emerging properties. It reproduces the responses of complex ESMs, for all aspects of the Earth system. We observe some quantitative differences with these models, most of them being due to the observational constraints. Some specific features of OSCAR also contribute to these differences, such as its fully interactive atmospheric chemistry and endogenous calculations of biomass burning, wetlands $CH_4$ and permafrost $CH_4$ and $CO_2$ 25 emissions. The main points of improvements are a low sensitivity of the land carbon cycle to climate change, an unstability of the ocean carbon cycle, the seemingly too simple climate module, and the too strong climate feedback involving short-lived species. Beyond providing a key diagnosis of the OSCAR model in the context of the reduced-complexity models intercomparison project (RCMIP), this work is also meant to help with the upcoming calibration of OSCAR on CMIP6 results, and to provide a large group of CMIP6 simulations run consistently within a probabilistic framework.



## 1. Introduction

Complex models such as Earth system models (ESMs) are used for climate projections (Collins et al., 2013). ESMs provide gridded detailed process-based outputs (Flato et al., 2013), but these strengths are mitigated by heavy computational costs. As a complement, some reduced-complexity models, also called simple climate models (SCMs), prove useful to investigate couplings and uncertainties (Nicholls et al., 2020; Clarke et al., 2014), especially for large ensembles of scenarios and statistical analysis of uncertainties to model parameters (Gasser et al., 2015; Li et al., 2016; Quilcaille et al., 2018). SCMs run significantly faster, thanks to a parametric modelling approach often calibrated on more complex models such as ESMs (Gasser et al., 2017; Meinshausen et al., 2011a; Hartin et al., 2015; Smith et al., 2018; Crichton et al., 2014; Dorheim et al., 2021). Although more simple than ESMs, those models exhibit a diversity in their modelling and calibration (Nicholls et al., 2020; Nicholls et al., 2021). In spite of this relative simple modelling approach, reduced complexity models still need to be validated. Even a very simple emulator may have difficulties to grasp some features of the Earth system, this diagnose would help in identifying what this emulator would be good at. Besides, reduced complexity models are often built as a coupling of modules, dedicated to aspects of the Earth system, such as the atmospheric chemistry, the oceanic carbon cycle, the climate response to radiative forcings, etc. These models may be calibrated as an emulator, with all modules calibrated together, for instance to emulate a single ESM. These models may also be calibrated as a combination of emulators, with each module calibrated separately, and this is the case of OSCAR. Under such an approach, each parametrization may be an existing ESM or an unforeseen combination. It broadens the range of modelling, but increase the need for validation. For these reasons, all reduced complexity models need to be validated in spite of their calibration and their relative simplicity.

In this paper, experiments designed under the Coupled Model Intercomparison Project 6 (Eyring et al., 2016) are used to diagnose the performances of the latest version of OSCAR, comparing its results to observations and other model outputs. We briefly describe the model and its update, the probabilistic setup used, and how it has been constrained using observations. We present the CMIP6 simulations run with OSCAR, and compare their results to the available CMIP6 ESM runs. Beyond diagnosis and despite being a simple model, OSCAR has a number of specificities that make it interesting to some of CMIP6-endorsed MIPs: CDRMIP (Keller et al., 2018) and ZECMIP (Jones et al., 2019) thanks to its advanced carbon cycle, and LUMIP (Lawrence et al., 2016) thanks to its book-keeping land use module (section 2.1). OSCAR is also part of the RCMIP project phases 1 and 2 (Nicholls et al., 2020; Nicholls et al., 2021), whose objective is to compare reduced complexity models together and against some CMIP6 and CMIP5 simulations.

Over the course of this study, we focus on several aspects of the model. To begin with, the approach based on the exclusion of diverging parametrizations and observational constraints is only briefly analyzed, for it is the one used in RCMIP phase 2 (Nicholls et al., 2021). Then, we use experiments from the DECK (Eyring et al., 2016) to simulate climate change over the historical period. Idealized experiments from the DECK and RCMIP (Nicholls et al., 2020) are used to evaluate the climate response, while other idealized experiments from the DECK and C4MIP (Jones et al., 2016) to evaluate the carbon cycle response. Some insights are obtained on the response to the solar geoengineering using GeoMIP experiments (Kravitz et al.,





2015), on carbon geoengineering using CDRMIP (Keller et al., 2018), and on the zero emission commited warming using ZECMIP (Jones et al., 2019). Experiments from DAMIP (Gillett et al., 2016), AerChemMIP (Collins et al., 2017), C4MIP (Jones et al., 2016) and LUMIP (Lawrence et al., 2016) form the basis for an attribution exercice of historical global temperature change. Land-use is investigated using LUMIP experiments. Climate projections are then obtained using ScenarioMIP (O'neill et al., 2016) and RCMIP experiments, with some variants of these scenarios in different MIPs.


## 2.    Experimental setup

### 2.1.   Brief description of OSCAR v3.1

OSCAR v3.1 is an open-source Earth system model of reduced complexity, whose modules mimic models of higher complexity, and meant to be used in a probabilistic fashion (Gasser et al., 2017). OSCAR v2.2 was entirely described in (Gasser
et al., 2017), and changes between v2.2 and v3.1 are summarized in (Gasser et al., 2020). We pinpoint that v3.1 is still calibrated on CMIP5 ESMs, then not meant to emulate CMIP6 models. Furthermore, each module is calibrated on available models, but not all ESMs have implemented every aspect modelled in OSCAR, such as permafrost or biomass burning. It means that OSCAR does not emulate the ESMs as such, but these models coupled with more modules.

Global surface temperature changes in response to radiative forcing follows a two-box model formulation (Geoffroy et al.,
2013b). Global precipitation is deduced from global surface temperature and the atmospheric fraction of radiative forcing (Shine et al., 2015). Linear scaling on the global variables is used to estimate regional temperature and precipitation changes, over five broad world regions (Iiasa, 2018b). OSCAR calculates the radiative forcing caused by greenhouse gases ($CO_2$, $CH_4$, $N_2O$, 37 halogenated compounds), short-lived climate forcers (tropospheric and stratospheric ozone, stratospheric water vapour, nitrates, sulphates, black carbon, primary and secondary organic aerosols) and changes in surface albedo.

The ocean carbon cycle is based on the mixed-layer response function of (Joos et al., 1996), albeit with an added stratification of the upper ocean derived from CMIP5 (Arora et al., 2013b) and with an updated carbonate chemistry. The land carbon cycle is divided into five biomes and the same five regions as previously, and each of the 25 biome/region combinations follows a three-box model (soil, litter and vegetation) described by (Gasser et al., 2020). The preindustrial state of the land carbon cycle is calibrated against TRENDYv7 (Le Quéré et al., 2018a) and its transient response to $CO_2$ and climate is
calibrated against CMIP5 models (Arora et al., 2013b).

Additionally, OSCAR endogenously estimates some key aspects of the carbon cycle. A dedicated book-keeping module tracks land cover change, wood harvest and shifting cultivation, which allows OSCAR to estimate its own $CO_2$ emissions from land-use change (Gasser et al., 2020; Gasser and Ciais, 2013). Permafrost thaw and the resulting emissions of $CO_2$ and $CH_4$ are also accounted for (Gasser et al., 2018). $CH_4$ emissions from wetlands are calibrated on WETCHIMP (Melton et al., 2013).
In addition, biomass burning emissions are calculated endogenously on the basis of the book-keeping module and wildfires



that are simulated as part of the land carbon cycle (Gasser et al., 2017). The latter emissions were subtracted from the input data used to drive OSCAR to avoid double counting.

The atmospheric lifetimes of non-$CO_2$ greenhouse gases are impacted by non-linear tropospheric (Holmes et al., 2013) and stratospheric (Prather et al., 2015) chemistries. Tropospheric ozone follows the formulation by (Ehhalt et al., 2001) but

recalibrated on ACCMIP (Stevenson et al., 2013). Stratospheric ozone is derived from (Newman et al., 2007) and (Ravishankara et al., 2009). Aerosol-radiation interactions are based on CMIP5 and AeroCom2 (Myhre et al., 2013), while aerosol-cloud interactions depend on the hydrophilic fraction of each aerosol and follows a logarithmic formulation (Hansen et al., 2005; Stevens, 2015). Surface albedo change induced by land-cover change follows (Bright and Kvalevåg, 2013). The impact of black carbon deposition on snow albedo is calibrated on ACCMIP globally (Lee et al., 2013) and regionalized

following (Reddy and Boucher, 2007).

### 2.2. CMIP6 and RCMIP experiments

A total of 99 experiments were run with OSCAR, 75 being from CMIP6 and 24 from RCMIP. A list of these experiments is provided in Table 1. We selected the experiments according to several criteria: typically, experiments are global and/or with

long time-series of output requested, and experiments do not overly focus on a given process or short time scales. In addition, RCMIP requested additional experiments to complement those of CMIP6, mostly extended and additional scenarios, including the RCP scenarios from the previous CMIP5 exercise (Meinshausen et al., 2011b). Between the CMIP5 and CMIP6 historical simulations, the concentration- and emission-driven ones, and the land-only experiments of LUMIP, eight different spin-up and control experiments had to be performed. Every spin-up is a recycling of the preindustrial forcing over 1000 years,.

We use driving datasets for historical concentrations of greenhouse gases (Meinshausen et al., 2017), projected concentrations of greenhouse gases (Esgf, 2018), emissions (Iiasa, 2018a; Gidden et al., 2019; Hoesly et al., 2018), land-use (Luh2, 2018), solar activity (Matthes et al., 2017), volcanic activity (Zanchettin et al., 2016) and the land-only climate climatology for LUMIP experiments (Lawrence et al., 2016). The extensions of scenarios are not those that were initially foreseen (O'neill et al., 2016), but those that have effectively been used during the CMIP6 exercise (Meinshausen et al., 2019).

The volcanic aerosol optical depth has been treated to scale and extend AR5 volcanic radiative forcing (Ipcc, 2013), to comply with the requirement of OSCAR to have a radiative forcing as driver for this contribution.

Every single experiment is run for 10,000 different configurations of OSCAR, drawn randomly from the pool of all possible parameters values in a Monte-Carlo setup (Gasser et al., 2017). Altogether, the combined experiments and Monte Carlo members sum to 569,700,000 simulated years.




### 2.3. Post-processing: exclusion and constraining

As described in (Gasser et al., 2017), most of the equations of OSCAR may use different sets of parameters or even different forms of equations. These parameters may arise from the training over different models, while the forms of equations may find their justification in the literature. Each combination of parameters and equations is defined as a configuration of OSCAR, and represent a different modelling of the Earth system. A Monte Carlo setup can be used with OSCAR over these configurations. This method for the uncertainty in the modelling of the Earth system comes with a side-effect: some combinations may be physically unrealistic. Other parameterizations may become numerically unstable when the model is pushed to the edge of the validity domain of its parametrizations. Therefore, the raw outputs of the simulations undergo two rounds of post-processing: one to exclude the diverging simulations, and one to constrain the resulting Monte Carlo ensemble.

In the exclusion round, we identify and discard the configurations that lead to a numerical divergence of the model. We developed a heuristic criterion to exclude these diverging runs by trial-error. Every experiment undergoes a thorough search for divergences. The 1118 configurations not causing any divergences in all the experiments are kept as a common set of configuratins for all experiments. More precisely, the criteria to identify divergences are as follows. We observe that when a divergence occurs, the oceanic carbon sink drops and then oscillates in most cases. An ocean sink larger than 20 PgC/yr in absolute value marks the configuration for exclusion. Some parametrizations under high warming scenarios exhibit an additional mode, not diverging in the strictest sense, yet, with the ocean carbon sink becoming a source and then switching back to a sink, which we identified as a physically unrealistic behaviour of the parametrization. To avoid this mode, we exclude configurations for which the ocean sink of the *ssp585*, *ssp370* and *1pctCO2* families have values out of the 0-20 PgC/yr domain. The identification of divergent configurations was improved by extending the 20 PgC/yr criteria to the land sink, the $CO_2$ emissions from LUC and the $CO_2$ emissions from permafrost, to ensure that the whole carbon cycle remains within reasonable boundaries. For the scenarios with an abrupt multiplication of the atmospheric concentration of $CO_2$, these criteria are applied over the last 50 years of the experiment only.

The need for exclusion is stronger as the atmospheric concentration of $CO_2$ and the global surface temperature increase. Most of the configurations are excluded thanks to the *ssp585*, *ssp370*, *1pctCO2* and *abrupt-4xCO2*. We acknowledge that when a significant fraction of the configurations is excluded, confidence in our model's result is lowered, but such a limitation of the validity domain is inherent to reduced-complexity models. We observed that in most cases, the reason of the exclusion is due to a diverging ocean sink. Increasing the number of sub-timesteps in the oceanic carbon module avoids this issue for a fraction of the configurations at the expense of the computational cost of the model. The ocean carbon cycle of OSCAR is its oldest module (Gasser et al., 2017), and should be redesigned for more stable behaviour under high-warming scenarios.

After this exclusion, the outputs of OSCAR are constrained using observations. The same method, exclusions and constraining, was used for the contribution of OSCAR in RCMIP phase 2 (Nicholls et al., 2021). The objective of this constraining round is to use the flexibility and the probabilistic frameworks of the reduced complexity models to synthesize


lines of evidence with the modelling of the Earth system. With OSCAR, we assess the physical likelihood of the model's

configurations using lines of evidence from the literature. To provide information on the climate system, the surface air ocean blended temperature change over 2000-2019 with reference to 1961-1990 are used, provided as an assessed range by RCMIP (Nicholls et al., 2021) from the HadCRUT 4.6.0.0 dataset (Morice et al., 2012). To constrain the carbon cycle, the CMIP5 cumulative compatible fossil fuel emissions over the concentrations-driven historical and 4 RCPs are used (Ciais et al., 2013b). For now, the OSCAR v3.1 model is calibrated on CMIP5, which motivates the use of these compatible emissions, and not

those of CMIP6. Besides, an initial set of constraints based solely on observations had revealed that using projections helped the overall constraining round, thanks to the larger perturbation in the scenarios than in the historical period. To further constrain the partitioning of the carbon sinks between land and ocean, we use data on the cumulative net ocean to atmosphere flux of $CO_2$ over 1750-2011 (Ciais et al., 2013b).

     For every constraint, we extend a method already used with OSCAR but with only one constraint (Gasser et al., 2020; Le

Quéré et al., 2018b). We assume a distribution from which we derive the likelikood of every configuration, as illustrated in equation A1 of (Gasser et al., 2020). The product of the probabilities over the set of constraints is the final likelihood of the configurations. Figure 1 illustrates the impact of these constraining steps. While the constraint for the surface air-ocean blended temperature change of 2000-2019 with reference to 1961-1990 is 0.54 K with a 90% confidence interval of 0.46-0.61 K, the model returns after constraining 0.55 K with a 90% confidence interval of 0.48-0.62 K. Applying this constraint successfully

reproduces the observed distribution, but also reduces the range in the other constraints, such as the cumulative net ocean carbon flux over 1750-2011 (Figure 1). The constraints on cumulative compatible emissions mostly impacts RCP6.0 and RCP8.5, transforming the bimodal distribution of the unconstrained OSCAR into a monomodal distribution. All final outputs and results are provided as the resulting weighted means and standard deviations, using the normalised likelihood as weight. The effect of this constraining is further discussed in the next sections.

**3.   Historical simulations and constraints**

     **3.1.  Effect of the constraints**

     Our constraining approach markedly corrects natural biases in OSCAR, as illustrated in Figure 1 for a few key outputs of the concentration-driven *historical* experiment. The change in global surface air temperature (GSAT) over 2000-2019 with regard to 1961-1990 is constrained to a value of $0.55 \pm 0.04$ K, instead of $0.60 \pm 0.11$ K without the constraint. Due to the

combination of observational constraints, the mean value remains still slightly larger than the constraint itself $0.54 \pm 0.05$ K, but the uncertainty range is significantly reduced to that of the observation.

     Regarding the carbon cycle, the unconstrained OSCAR shows a negative bias in the cumulative net land carbon sink (i.e. a too weak removal), balanced by lower cumulative compatible fossil-fuel emissions. Using observational constraints reduces these biases but does not entirely remove them. After applying the constraints, the uncertainty ranges of the net land flux and

of fossil-fuel emissions are significantly reduced. The ocean carbon sink over 1750-2011 of the unconstrained OSCAR is 159





± 20 PgC, higher than the one of IPCC AR5 (Ciais et al., 2013b), 155 ± 18 PgC, in terms of mean and standard deviation. Using this constrain, the mean of OSCAR is increased and the range decreased, reaching 163 ± 15 PgC.

### 3.2. Concentration- and emission-driven historical simulations

The concentration- and emission-driven historical experiments (i.e. *historical* and *esm-hist*, respectively) were run with

OSCAR. Their forcers differ only on $CO_2$: the atmospheric $CO_2$ is prescribed in the former, whereas in the latter, fossil-fuel emissions are prescribed and atmospheric $CO_2$ is fully interactive. In the concentrations-driven *historical*, compatible fossil-fuel emissions are back-calculated after the simulation (Jones et al., 2013; Gasser et al., 2015).

Altogether, these two simulations are relatively close, as shown in Figure 2, but with noticeable differences.

Looking at the carbon-cycle variables, we observe that up to the 1940s, *esm-hist* is relatively similar to *historical* in terms

of fossil-fuel $CO_2$ emissions, atmospheric $CO_2$ and both carbon sinks. The difference observed afterwards can essentially be explained by the fact that the emission-driven simulation entirely misses the 1940s plateau in atmospheric $CO_2$. Such a miss is typical of ESMs (Bastos et al., 2016). For comparison after 1959, we use data from the Global Carbon Budget (Friedlingstein et al., 2020) whose assessed ocean carbon sink is slightly closer to our *historical* than to our *esm-hist*. The net carbon flux from atmosphere to land (i.e. the aggregate of the land sink, emissions from LUC, and those from permafrost) of the two historical

experiments are similar from the 1980s onward.

Looking at the effective radiative forcings (ERF), that of $CO_2$ in the concentration-driven *historical* is directly deduced from the prescribed $CO_2$ atmospheric concentration (Meinshausen et al., 2017), but slightly higher by about 0.1 W.m$^{-2}$ than the central value from the 5[th] Assessment Report (AR5) (Myhre et al., 2013). The central value from AR5 (1.82 W.m$^{-2}$) is calculated with reference to 1750, but becomes 1.66 W.m$^{-2}$ when calculated with reference to 1850. Because of changes mostly

in the $CO_2$ concentration in 1850 in CMIP6 data, this value increases to 1.70 W.m$^{-2}$. With OSCAR and prescribed $CO_2$ emissions, the atmospheric $CO_2$ in *esm-hist* is higher than in *historical*, the ERF of $CO_2$ is 0.2 W.m$^{-2}$ higher than in the AR5. The ERF of other greenhouse gases are consistent with (Myhre et al., 2013). For most ERF components, there is very little difference between *historical* and *esm-hist*. OSCAR's overall ability to simulate the RF of short-lived species compares well with the IPCC AR5 values. Contributions to the warming from aerosols and ozone are consistent as well, although OSCAR

tends to amplify these contributions. It may be caused by overestimated biomass burning emissions, and this will be examined more in-depth in a future analysis. Since these biases were already diagnosed in the description paper of OSCAR (Gasser et al., 2017), it shows that our constraining does not markedly alter these aspects of the model. Additional constraining could be introduced for separate RF components, albeit this would likely weaken the efficiency of existing constraints.

Looking at climate variables, the increase in GSAT in both historical experiments are consistent with the Special Report

on Global Warming of 1.5C (Ipcc, 2018) and with the historical reconstruction by (Cowtan and Way, 2013). During the choice of constraints (sections 2.3 and 3.1), we observed that constraints on temperatures impact much more our results than the other type of constraints. Even while the set of constraints is expanded, constraints on temperature have a lasting influence over all outputs. The *esm-hist* simulation shows a higher GSAT, and appears to be further away from the observations. This is mostly



the result of the higher atmospheric $CO_2$ seen earlier, and it suggests a different set of constraining weights could be used for

the emission-driven runs. We choose not to, for the sake of consistency. Comparing the effective radiative forcing (ERF) of OSCAR to the one of the IPCC AR5 (Myhre et al., 2013), we note differences caused by volcanic eruptions. Beyond the update of the time-series of volcanic activity itself, OSCAR make use of of a warming efficacy of 0.6 for stratospheric volcanic aerosols (Gasser et al., 2017; Gregory et al., 2016). Finally, the total ocean heat content is well reconstructed, although the range of OSCAR is larger than the observed one (Von Schuckmann et al., 2020), suggesting this could also be considered a

potential constraint for the model in future work.

## 4. Idealised experiments

### 4.1. Climate response (DECK & RCMIP)

Simulations with an abrupt increase in atmospheric $CO_2$ (and thus in radiative forcing) are typically used to diagnose the climate response of complex models. We use three such experiments from CMIP6 and RCMIP with quadrupled, doubling and

halving atmospheric $CO_2$ (*abrupt-4xCO2*, *abrupt-2xCO2* and *abrupt-0p5xCO2*). These experiments can be used to estimate the ECS of an ESM or a model such as OSCAR (Gregory et al., 2004) and investigate how this metric is influenced by the intensity of the forcing. These results are shown in Figure 3.

The ECS is defined as the equilibrium temperature that results from the doubling of the preindustrial atmospheric concentration of $CO_2$ (Gregory et al., 2004). The ECS and its calculations have evolved with the integration of new components

to climate models (Meehl et al., 2020). In regard of the computational cost of the ESMs, reaching this equilibrium takes a time long enough to use Gregory's method (Gregory et al., 2004) to calculate the ECS or alternative methods (Lurton et al., 2020; Schlund et al., 2020). The ECS using the Gregory method is actually not exactly the equilibrium climat sensitivity per se, but rather an "effective climate sensitivity" (Sherwood et al., 2020). Paleoclimate data shows that feedbacks from vegetation, biogeochemistry or dust affect the equilibrium (Friedrich et al., 2016; Rohling et al., 2012). From CMIP5 to CMIP6, some

ESMs have improved their treatment of the biogeochemistry and the vegetation, leading to alteration in feedbacks and aerosols fields (Meehl et al., 2020). This evolution participates in the observed changes in ECS from CMIP5 to CMIP6, attributed to cloud effects (Zelinka et al., 2020) and the pattern effect (Dong et al., 2020).

In OSCAR, there are two ways of estimating the ECS. First, because OSCAR is not process-based, the ECS is actually a parameter of the model. Since the formulation of the climate module is linear (Gasser et al., 2017; Geoffroy et al., 2013b), we

also know that this value is independent of the intensity of the abrupt experiment. This parameter was calibrated on the *abrupt-4xCO2* experiment run by CMIP5 models and normalised to OSCAR's estimate of RF for a quadrupling of $CO_2$ (Gasser et al., 2017). Under this definition, the ECS of OSCAR follows the Gregory's method and does not account for all feedbacks of OSCAR. When using parameters from OSCAR, the climate feedbacks actually included in the estimated ECS depend on the CMIP5 models used for calibration. If calibrated on general circulation models (GCMs), only the so-called Charney feedbacks

are included (i.e. Planck, water vapour, lapse rate, sea-ice albedo, and clouds) with the possible addition of the $CO_2$





physiological feedback (Sellers et al., 1996). However, when calibrated on ESMs, additional feedbacks pertaining to interactive biogeochemical cycles may be included, depending on what exact processes are implemented in a given ESM. The second way of estimating the ECS in OSCAR is to define it as the GSAT change at the end of the 1,000 years of the abrupt experiments. Here, all of the feedbacks integrated in OSCAR are accounted, for instance those about biogeochemistry.


Values related to these two approaches are presented in Table 2. The ECS calculated using parameters of OSCAR, hence comparable to Gregory's approach, is $2.78 \pm 0.47$ K when constrained, while the unconstrained one is $3.17 \pm 0.63$ K. This, by construction, is consistent with the AR5 estimates (Collins et al., 2013), but also with more recent assessments (Gregory et al., 2020). Because we use observational constraints, these results are lower than the CMIP5 range $2.1 - 4.7$ K (Andrews et al.,

2012). The CMIP6 range, $1.8 - 5.6$ K (Zelinka et al., 2020; Meehl et al., 2020) is even higher than the CMIP5 range. The higher values for the ECS from some CMIP6 models are significantly reduced when constraining (Nijsse et al., 2020; Bonnet et al., 2021), with ECS even lower – 1.38K with a likely range of 1.3-2.1K – than those shown by OSCAR here. Overall, these values provided by OSCAR remain consistent with the litterature (Sherwood et al., 2020). Similarly, the TCR and the TCRE of the unconstrained OSCAR are consistent with the CMIP5 values (Meehl et al., 2020) and (Gillett et al., 2013), thanks to the

calibration of the ECS in OSCAR. Constraining OSCAR reduces all these metrics both in value and in range, and we attribute this effect to the constraint on historical warming. This reduction effect is similar to what was shown recently for CMIP6 models (Tokarska et al., 2020).

The abrupt non-doubling experiments are rescaled using the total radiative forcing of $CO_2$. This approach is illustrated in Figure 3, and it leads to an unconstrained ECS of $2.74 \pm 0.52$ K (Table 2), reduced to $2.52 \pm 0.33$ K with the constraints.

Overall, the ECS is remarkably consistent in terms of average, standard deviation and even skewness across the three step experiments. This is due to the construction of OSCAR, with a prescribed logarithmic dependency of the radiative forcing of $CO_2$ to its atmospheric concentration (Lurton et al., 2020). This ECS is lower than with the first approach, because it includes several Earth system feedbacks related to short-lived species that are left free to change during the simulations, owing to the experimental protocol. In OSCAR, this is mostly explained by an increase in the atmospheric load of tropospheric aerosols

(and ozone) caused by the endogenous emission of precursors through biomass burning. These feedbacks are also illustrated in Figure 3. The RF resulting from the prescribed change in atmospheric $CO_2$ ($7.42$ W.m$^{-2}$ under quadrupled $CO_2$) is partially compensated by short-lived climate forcers. In the case of *abrupt-4xCO2*, the RF sums up to $3.46 \pm 0.25$ W.m$^{-2}$, because of a cooling by scattering aerosols ($-0.21 \pm 0.16$ W.m$^{-2}$) and aerosol-cloud effects ($-0.21 \pm 0.15$ W.m$^{-2}$), besides an additional warming from absorbing aerosols ($0.13 \pm 0.08$ W.m$^{-2}$). Finally, from Table 2, we note that contraining reduces the parameter-

based ECS by 0.44 K, while the one with all feedbacks has its ECS reduced by 0.22 K. It means that the feedbacks appearing only in the second approach, which are mostly related to biogeochemistry, are strongly impacted by the constraints than the others.



### 4.2. Carbon-cycle response (DECK & C4MIP)

The *1pctCO2* experiment, in which atmospheric $CO_2$ increases by +1% every year, is part of the DECK. Two variants of *1pctCO2* have been performed as part of the C4MIP exercise (Figure 4). In *1pctCO2-rad*, atmospheric $CO_2$ only has a radiative effect on the climate system, as a preindustrial level of $CO_2$ is seen by the carbon cycle. In *1pctCO2-bgc*, only the carbon cycle is affected by $CO_2$, whereas a preindustrial $CO_2$ is prescribed to the climate system. The outputs of OSCAR v3.1 on these experiments are consistent with past C4MIP results (Arora et al., 2013b). The global mean surface temperature responds about

linearly to the exponential increase in $CO_2$, because of the implemented logarithmic dependency of the radiative forcing of $CO_2$ to its atmospheric concentration. Carbon sinks rise in response to the increase in atmospheric $CO_2$, but the resulting warming dampens the sinks.

These three experiments can be used to calculate the carbon-concentration and carbon-climate feedback metrics, respectively $\beta$ and $\gamma$. These metrics, defined and used in former C4MIP exercises (Arora et al., 2013b; Friedlingstein et al.,

2006; Arora et al., 2019), are a means to diagnose the model's sensitivities of the carbon stocks in the land and in the ocean to changes in atmospheric $CO_2$ or GSAT. Table 3 summarizes these results. As explained by (Arora et al., 2013b), there are three methods to combine the three experiments to calculate the metrics: subtracting *1pctCO2-bgc* from *1pctCO2-rad* (noted R-B, hereafter), subtracting *1pctCO2* from *1pctCO2-bgc* (B-F), and subtracting *1pctCO2* from *1pctCO2-rad* (R-F). Methods R-B and B-F are almost equivalent for $\beta$, while methods R-B and R-F are almost equivalent for $\gamma$. Although LUC affects these

metrics (Melnikova et al., 2021), these experiments are designed to have a constant LUC.

Table 3 shows that $\beta$ under the R-F method are lower than the R-B and B-F, because the non-linearity of the Earth system reduces the sensitivity of land and ocean carbon to atmospheric $CO_2$. Similarly, $\gamma$ under the R-B and R-F are higher than under the B-F, but the non-linearity here is added to R-B and B-F (Arora et al., 2013a). Applying our observational constraints increases the absolute values of $\beta_{land}$ and $\gamma_{land}$ of OSCAR, but it does not affect significantly the $\beta_{ocean}$ and $\gamma_{ocean}$. The only

exception is the $\gamma_{ocean}$ under the method B-F. We note that the unconstrained OSCAR v3.1 is closer to the CMIP5 exercices, be it at 2x or 4xCO2. This result can be explained with OSCAR v3.1 being calibrated on CMIP5. However, the unconstrained $\beta_{land}$ is the only one to be closer to CMIP6 than to CMIP5. The cause of this difference in the $\beta_{land}$ remains unclear, but may come from the form of equation for the fertilization effect. The configurations of OSCAR are not only different parameters, but also different equations. Here, half of the configurations of OSCAR follow a logarithmic formulation of the fertilisation

effect (Gasser et al., 2017), which may not be convex enough to properly represent a saturation effect. We note that in our assessment, the land includes permafrost carbon, which was not the case in CMIP5 assessment, but the permafrost is mostly sensitive to increase in temperatures, the only sensitivity to atmospheric $CO_2$ is due to non-linear contributions.

Overall, the unconstrained carbon cycle of OSCAR v3.1 is well in line with CMIP exercices, particulary CMIP5. Yet, the sensitivity of the oceanic carbon stock to increase in GSAT remains too high. This bias in the ocean module could be attributed

to the stratification effect introduced in v2.2 (Gasser et al., 2017). In any case, this suggests that our carbon cycle may be too optimistic, which will clearly appear in our emission-driven simulations.





### 4.3. Solar geoengineering (GeoMIP)

Experiments of GeoMIP (Kravitz et al., 2015) are designed to investigate the geoengineering techniques of Solar Radiation Management (SRM). Although OSCAR is not suited for all GeoMIP experiments, as it lacks any spatially resolved process, a

few simulations remained accessible to our model. We run experiments *G1* and *G2*: G1 essentially follows *abrupt-4xCO2*, albeit with a changed incoming solar radiation that compensates for the radiative forcing caused by the increasing atmospheric $CO_2$ For *G2*, an identical principle is applied but using *1pctCO2* as a basis. As explained by (Kravitz et al., 2011), the change in solar radiation compensates solely for the radiative forcing of $CO_2$. However, it does not compensate for other radiative effects introduced by biogeochemical feedbacks, such as the fertilization by $CO_2$, affecting the carbon cycle, thus changing

biomass burning emissions. Figure 5 shows that offsetting the $CO_2$ radiative forcing with a change in solar activity effectively compensates the change in GSAT. However, we simulate that the GSAT decreases in *G1* and *G2* to reach $-0.08 \pm 0.20$ K and $-0.07 \pm 0.20$ K, respectively, at the end of simulations. The compensation of the sole radiative forcing of $CO_2$ does not balance other feedbacks. There remains an additional radiative forcing, mostly due to changes in aerosols (as also shown in Figure 3), which results in this relatively small cooling in *G1* and *G2*. We estimate that in OSCAR about half of this effect is caused by

the vegetation being fertilized by $CO_2$ and fuelling increased natural biomass burning emissions, and the remaining half is caused by the direct impact of GSAT on the atmospheric lifetime of aerosols (not shown). We note that the latter effect could be poorly estimated, in these specific experiments, as OSCAR's formulation for the lifetime of aerosols depends only on GSAT and not on the precipitation intensity.

Indeed, global precipitation does not respond in a similar way, because changes in atmospheric $CO_2$ and solar radiation

have a different impact of the hydrological cycle (Andrews et al., 2010). In spite of a fully compensated GSAT change, global precipitation is significantly reduced in *G1* and *G2*, showing that such SRM technique does not entirely negate climate change. This demonstrates that OSCAR is capable of reproducing this well-established effect of this SRM technique (Boucher et al., 2013). One added value of having a fully coupled ESM run these GeoMIP experiments is that we can also provide an estimate of the impact of the SRM technique on the carbon cycle. Figure 5 also shows that the land and ocean carbon stocks are increased

in *G1* and *G2*, respectively by about 33% and 20% at the end of the simulations, owing to the loss of carbon sink efficiency that is avoided by maintaining the temperature to its preindustrial level.

### 4.4. Carbon geoengineering (CDRMIP)

Experiments of CDRMIP are designed to investigate the consequences of carbon dioxide removal for the Earth system (Keller et al., 2017). In *1pctCO2-cdr*, the atmospheric $CO_2$ increases by 1% every year (just like *1pctCO2*), but after 140 years,

the atmospheric $CO_2$ decreases following a pathway at the same rath than the ramp-up period. Once $CO_2$ has returned to its preindustrial state, the experiment is extended over 1000 years. As shown in Figure 6, the GSAT reaches $3.68 \pm 0.39$ K at the end of the ramp-up forcing, and it goes back to $0.85 \pm 0.22$ K at the end of the ramp-down forcing. For all variables, such as the $CH_4$ emissions from wetlands, removing $CO_2$ from the atmosphere during ramp-down effectively reduces the perturbation





in the variable that was induced by the ramp-up, albeit within a different time frame that is typical of a dynamic hysteresis
(Boucher et al., 2012). Even the permafrost carbon stock slowly reconstitutes itself, once the global temperature change is
sufficiently reduced. However, the whole Earth system is not fully recovered as soon as the preindustrial level of atmospheric
$CO_2$ is reached. To return within 10% of the maximum perturbation at the end of the $CO_2$ ramp-up, it takes GSAT an average
110 extra years, and the land carbon stock an average 26 years. At the end of the 1000-year extension, the oceanic carbon stock
remains at about 19% of its maximum perturbation.

360 Other CDRMIP experiments based on pulses of carbon emission or removal in an emission-driven configuration were
performed to evaluate the response of the Earth system to CDR. These experiments are used to calculate the Absolute Global
Warming and Temperature Potentials (AGWPs and AGTPs) of $CO_2$, which serves to establish the Global Warming and
Temperature Potentials (GWPs and GTPs) of other greenhouse gases (Myhre et al., 2013). In *esm-pi-CO2pulse*, a 100 PgC
pulse is emitted from the preindustrial environmental condition in 1860, whereas 100 PgC are removed in *esm-pi-cdr-pulse*.
365 In *esm-yr2010CO2-CO2pulse*, the 100 PgC pulse is applied in 2015 but under 2010 environmental conditions, whereas these
100 PgC are removed at the same date in *esm-yr2010CO2-cdr-pulse*. We calculate timeseries of AGWPs and AGTPs under
these experiments (Figure 7), and note how close they are. We pinpoint that, just like the other experiments, we are calculating
these potentials with the interactive permafrost of OSCAR. The larger source of differences lies in the background: under
preindustrial environmental conditions, emission pulses have a stronger AGWP or AGTP over the short term, but this is
inverted over the longer term. Over the short term, this is due to the logarithmic expression of the $CO_2$ radiative forcing that is
less saturated under preindustrial conditions. Over the long term, this is due to the deterioration of the carbon sink capacities
under current conditions (Raupach et al., 2014). Similar reasons explain why a pulse of carbon removal cools the atmosphere
slightly more over the short term than a pulse of emission warms it, but less over the long term. Our results cannot be compared
to the final CDRMIP results yet, for they are unpublished, but they are consistent with those obtained with a model of
intermediate complexity (Zickfeld et al., 2021).

### 4.5. Zero-emission commitment (ZECMIP)

ZECMIP aims at investigating the zero-emission commitment (ZEC), that is the additional warming that follows a cessation
of anthropogenic $CO_2$ emissions (Jones et al., 2019). Two categories of experiments were performed. The first one (called
branched experiments) is a variation of the emission-driven *1pctCO2*, in which emissions cease once they reach 750 PgC,
1000 PgC or 2000 PgC of cumulative value. These different levels of emission are meant to evaluate the state dependency of
ZEC. The second category consists in three bell-shaped emission pathways whose cumulative emissions are the same as in the
branched experiments. This was proposed by ZECMIP to evaluate the dependency of the ZEC on $CO_2$ emission rate, as the
emission rate at the time of cessation is near 0 in these bell experiments (whereas it is very high in the branched ones).

Figure 8 shows the timeseries of the ZEC in both sets of experiments. Two features are remarkable. First, the ZEC in
branched experiments is systematically lower than the one in bell experiments. We attribute this response to the biomass
burning and aerosol lifetime feedbacks (the same that affect the ECS) whose response to temperature change happens within



the same year. And so, in the branched experiments, the abrupt cessation of $CO_2$ emissions triggers an abrupt response of temperature change that is amplified by the feedbacks. Conversely, in the bell experiments, since the cessation is smoother, no abrupt response is visible on the very short term. Second, the ZEC for a cumulative emission of 2000 PgC is much higher than

in the two other cases, highlighting a strong non-linearity in the model that we attribute to the permafrost response, in complete agreement with our previous work (Gasser et al., 2018). As illustrated in Table 4, OSCAR v3.1 estimates a ZEC (in the reference case of the *esm-1pct-brch-1000PgC* experiment) that is within the range of ZECMIP (Macdougall et al., 2020), although the long-term decrease seems to happen later in OSCAR.

## 5.  Attribution experiments (DAMIP, AerChemMIP, C4MIP & LUMIP)

DAMIP (Gillett et al., 2016) designed a number of experiments meant to attribute the observed climate change to anthropogenic and natural factors. Since OSCAR does not feature any internal variability, it cannot contribute to the "detection" part of DAMIP. However, with its 10,000 Monte Carlo elements, OSCAR is fully capable of carrying out the "attribution" part. To do this attribution, DAMIP relies on experiments that follow the *historical* one, but in which only one forcing or group of forcing is turned on. Conversely, a number of other MIPs introduced attribution experiments in which all

forcings but the ones studied are turned on. Neither of these approaches explicitly consider the non-linearities of the system, however. Other more robust methods of attribution to forcings exist (Trudinger and Enting, 2005) and have been used with OSCAR in the past (Gasser, 2014; Li et al., 2016; Fu et al., 2020; Ciais et al., 2013a) but we here focus on results made possible with the CMIP6 experiments and presented in Table 5.

In the *historical* experiment, we find a change in GSAT of 0.98 ± 0.17 K in 2006-2015 with regard to 1850-1900, which

is in line with observations because of our constraining setup. Of this total, we find only ~0.03 K was caused by natural forcings, of which ~0.02 and ~0.01 were caused by solar and volcanic activity, respectively. Note that our volcano-related forcing is defined against an average and constant volcanic activity during the preindustrial period, which explains the (slightly) positive response caused by this forcing over the recent past where no major volcanic eruption happened. In the IPCC terminology, our results lead to the conclusion that it is *extremely unlikely* (i.e. likelihood <1%) that natural factors alone are

causing the current observed climate change. This is of course consistent with the IPCC conclusions (Bindoff et al., 2013), and with more recent results as well (Gillett et al., 2021). Nevertheless, we note that our constraining reduces the uncertainty range of all simulations, including those driven only by natural forcings. For the simulations under natural forcings, the range from the constrained OSCAR is smaller than the ones from (Gillett et al., 2021), which may suggest an over-constraining. It may be solved by using different methods for constraining climate simulations (Nicholls et al., 2021).

Since DAMIP did not include an experiment in which only anthropogenic forcings would be turned on (i.e. natural forcings would be turned off), we cannot conclude as to the(Gillett et al., 2021) complementary probability of observed climate change being caused only by anthropogenic factors. Attribution to groups of anthropogenic forcings is possible, however. We find that 1.25 ± 0.11 K (corresponding to about 128 % of the recent warming) was caused by well-mixed greenhouse gases



(WMGHGs), -0.26 ± 0.22 K (-27 %) was by near-term climate forcers (NTCFs), and another -0.03 ± 0.03 K (-3 %) by land-
use change. Notably, the DAMIP experiment (*hist-aer*) and the AerChemMIP one (*hist-piNTCF*) led to very similar estimates
of the contribution of NTCFs (Table 5), which highlights that this part of our model behaves in a fairly linear fashion. Going
further in isolating individual forcings, we also estimate that $CO_2$ caused 0.74 ± 0.06 K, chlorofluorocarbons and hydro-
chlorofluorocarbons (i.e., CFCs and HCFCs) caused 0.13 ± 0.02 K, stratospheric $O_3$ caused -0.03 ± 0.03 K, and all aerosols
together caused -0.33 ± 0.21 K (including direct and indirect effects). We point out that details on $CH_4$, $N_2O$ or tropospheric
ozone cannot be provided, because of the lack of relevant CMIP6 experiments.

The extent to which this attribution to specific forcings is comparable to existing studies remains debatable. One notable
limitation of OSCAR, in this respect, is that the model's climate response is not forcing-dependent. The use of effective
radiative forcing is supposed to ensure that the temperature response to $CO_2$ and non-$CO_2$ forcings is similar, at least for the
long-term steady-state (Myhre et al., 2013). However, recent work has pointed out that the response may strongly depend on
the forcing agent (Marvel et al., 2016), thus casting a degree of doubt on our attribution results. More work to integrate such
differentiated responses in reduced-complexity models is warranted.

## 6.    Land-use experiments (LUMIP)

LUMIP consists of experiments specifically focusing on land-use activities, and most of them are run by the Earth system
models in a so-called "offline" fashion  (Lawrence et al., 2016). It means that a reconstruction of past climate variables GSWP3
(Lawrence et al., 2016; Van Den Hurk et al., 2016) is prescribed to the model, so that the land module is actually decoupled
from the rest of the model. Despite its simplicity, OSCAR has an added-value in running those simulations, as it embeds a
book-keeping module that endogenously estimates $CO_2$ emissions from land-use and land-cover change. The main land carbon
fluxes and stocks simulated under the reference experiment (dubbed *land-hist*) are shown in Figure 9, along with three sets of
sensitivity experiments described hereafter. The results are similar to those obtained recently with the same version of the
model but with slightly differing forcings and a different constraint (Gasser et al., 2020). The simulated land carbon stock
decrease up to the 1970s, because of land-use activities emitting more $CO_2$ than the sink absorbs thanks to $CO_2$ fertilization
and other factors. The carbon stock of 2010 is higher than the one of 1850 by only 1 ± 42 PgC. For comparison, the GCB 2020
provides for 1850-2014 a net budget for the land sink and $CO_2$ emissions from LUC of -5 ± 90 PgC (Friedlingstein et al.,
2020).

The experiments *land-cCO2* and *land-cClim* are used to disentangle the contribution of $CO_2$ fertilization and changing
climate on the land carbon cycle. In *land-cCO2*, the atmospheric $CO_2$ is constant and set to preindustrial value. In *land-cClim*,
the climate drivers loop over the year 1901-1920 of the data set, thus simulating a preindustrial climate. Figure 9 shows the
differences; for example, *land-hist - land-cCO2* illustrates the effect of atmospheric $CO_2$ on the variables of interest. Thanks
to these experiments, we show that $CO_2$ is the main driver of the land sink in OSCAR, driving most of the trend, with climate
bringing a significant interannual variability but virtually no trend, except over the recent past. In 2010, climate caused a small





difference of -10 ± 10 PgC in total land carbon stock, while $CO_2$ did one of 141 ± 42 PgC. This has to be balanced with the results of the C4MIP idealized experiments, however, where we saw OSCAR is less sensitive to climate change than CMIP5 models. Additionally, we see that the effect of climate and $CO_2$ on land-use and land-cover change emissions is minor, which is consistent with the fact that they are firstly determined by preindustrial carbon densities (Gasser et al., 2020; Gasser and
Ciais, 2013).

A second set of experiments is meant to investigate the impact of land-use practices. Land-cover change contributed -152 ± 44 PgC to the 2010 change in land carbon stock since 1850, which corresponds to most of the total land-use and land-cover change emissions. Notably, it also reduced the land sink – an effect called the loss of additional sink capacity that has been diagnosed and quantified with OSCAR in the past (Gasser et al., 2020; Le Quéré et al., 2018b; Gasser and Ciais, 2013;
Friedlingstein et al., 2020). Shifting cultivation (i.e. rapidly rotating land-use change between agriculture and natural ecosystems) had a relatively low impact on $CO_2$ emissions, leading to a change in land carbon stock of -8 ± 2 PgC at the end of the simulation in 2010. Similarly, wood harvest (in woody ecosystems that do not see land-cover change) had an overall impact of -16 ± 4 PgC. Both shifting cultivation and wood harvest have no impact at all on the land sink, by construction of their formulation in OSCAR (Gasser et al., 2020). Finally, the effect of having cropland-specific parameters in the model is
isolated thanks to the *land-crop-grass* experiment, in which new croplands are treated as grasslands. Having grasslands instead of croplands increases both the land sink and the $CO_2$ emissions from land-use and land-cover change, resulting in a land carbon stock higher by 31 ± 26 PgC. All these values are entirely in line with an existing assessment of those land-use practices in which an earlier version of OSCAR took part (Arneth et al., 2017).

The third set of experiments relates to varying input data sets of land-use and land-cover change drivers. Two of these
(*land-hist-altLu1* and *land-hist-alLu2*) relied on the two variations of the main LUH2 data set known as the "High" and "Low" variants (respectively) (Hurtt et al., 2020). We find that the so-called low variant leads to slightly higher land-use and land-cover change emissions amounting to an land carbon stock lower by 8 ± 2 PgC over the whole period. The high variant produces slightly lower total emissions, leading to a land carbon stock higher by 17 ± 5 PgC. Neither variant has a significant impact on the land sink. According to the description of these two variations (Hurtt et al., 2020), they differ from the default
data set mostly in the harvest of biomass, and are very similar from 1920 onwards. The last LUMIP experiment run with OSCAR is one that uses the primary data set but an alternative starting year (*land-hist-altStartYear*). This required making an additional spin-up of the model under the environmental conditions and land cover of year 1700. Compared to the reference experiment, we find a slightly higher land sink after 1850 that decreases through time, owing to the ecosystems not being at steady state at that date. Similarly, emissions are slightly higher but the difference to the reference case tends towards zero as
the legacy of land-use and land-cover change prior to 1850 fades away. The land carbon stock in 2010 is dominated by the increased land sink and amounts to a small increase of -17 ± 13 PgC in the land. Comparing the latter value with the total change in land carbon in the reference experiment suggests starting simulations in 1850 instead of 1700 or 1750 introduces a non-negligible bias in the CMIP6 exercise.



## 7. Climate change projections

### 7.1. Main CMIP6 scenarios (ScenarioMIP & RCMIP)

ScenarioMIP (O'neill et al., 2016) chose eight particular SSPs taken from the SSP scenario database (Riahi et al., 2017) to cover a range of socio-economic assumptions and climate targets, and then harmonised them to become the default CMIP6 scenarios to be run by ESMs (Gidden et al., 2019). ScenarioMIP mostly required concentration-driven simulations up to year 2100 or sometimes 2300, however, which was complemented in RCMIP by extending all scenarios up to 2500 and systematically running emission-driven simulations in addition (Nicholls et al., 2020). Figure 10 displays projections of some key global variables of the Earth system following these scenarios, and Table 6 focuses on projected GSAT changes.

The climate target dimension of the SSP scenarios is defined similarly to the RCPs' as the total RF targeted in 2100 (Van Vuuren et al., 2011). Table 6 shows that this targeted RF is overall within the 1-σ uncertainty range of all our concentrations-driven projections. In the cases with notable differences, such as *ssp460*, the actual RF reached by the reduced-complexity model MAGICC (Iiasa, 2018a) for this scenario is 5.29 W.m$^{-2}$, which is then in the range of OSCAR. Because MAGICC has been used for the design of these scenarios, it demonstrates that we remain consistent with the scenarios. Emission-driven SSPs show lower RF than their concentration-driven counterpart, which can be attributed to a low bias in the atmospheric $CO_2$ that is especially visible in high $CO_2$ scenarios. This bias is a result of our constraining approach that favoured configurations with strong $CO_2$-fertilization (as also seen with the C4MIP results). Under high $CO_2$ scenarios, this bias is likely worsened by our exclusion procedure during the post-processing, as very high $CO_2$ tends to make the model more unstable. The very low uncertainty range we obtain for projected atmospheric $CO_2$ in emission-driven simulations seems over-confident. However, we note that the constraints were derived using concentration-driven simulations (that are the focus of CMIP6), and so they may not apply properly to emission-driven simulations.

The constraining approach contributes in having the increases in GSAT shown in Table 6 for concentration-driven experiments to be lower than the CMIP6 models we could compare our results to. The uncertainty range simulated by OSCAR is also much lower, again owing to our constraining approach. With a relative uncertainty in GSAT change in 2500 of ±13% under the warmest scenario (SSP5-8.5), one may wonder whether these projections are over-constrained. This stems from our constraining of the climate response, as also shown by the relatively small uncertainty range in ECS in the idealised abrupt $CO_2$ experiments. Further developing that module by adding one or two key parameters (Geoffroy et al., 2013a; Bloch-Johnson et al., 2015) would provide more degrees of freedom and likely release part of the constraint. When projecting temperature change in an emission-driven mode, however, the uncertainty range is larger, because of the additional uncertainty related to the biogeochemical cycles.

The CMIP6 values are here computed from CMIP6 timeseries. However, some CMIP6 models exhibit higher warmings than in previous assessments, and observations can be used to constrain the future warming (Tokarska et al., 2020). Using their table S4, the warming in 2081-2100 with reference to 1995-2014 under SSP5-8.5 for the constrained CMIP6 models is 3.44 ± 0.67 K and 3.11 ± 0.36 K for OSCAR v3.1 constrained. For SSP1-2.6, the values are respectively 0.94 ± 0.30 K and 0.76 ±





0.17 K. Thus, the observational constraints that we have used contribute to explain the differences to the raw CMIP6 data. Nevertheless, the climate module of OSCAR v3.1 could still be improved.

### 7.2. Variant focused on NTCFs (AerChemMIP)

The *ssp370-lowNTCF* scenario is a variant of the *ssp370* differing by its lower emission of short-lived pollutants affecting the RF of NTCFs. As illustrated in Figure 11, the variant leads to a somewhat equivalent warming, although with very slightly less cooling from NTCFs. This almost negligible effect on global temperature is actually the result of two large but compensating effects that manifest the most between 2050 and 2100. The lower emission of warming NTCFs leads to absorbing aerosols (i.e. BC) warming less by $-0.21 \pm 0.11$ W m$^{-2}$ and tropospheric ozone warming less by $-0.21 \pm 0.03$ W m$^{-2}$

in 2100. Conversely, it also leads to scattering aerosols cooling less by $0.33 \pm 0.12$ W m$^{-2}$ and the indirect aerosol effects cooling less by $0.26 \pm 0.13$ W m$^{-2}$ at the same date. This results in a small increase of the total radiative forcing of $0.15 \pm 0.20$ W m$^{-2}$ and a GSAT change of only $0.07 \pm 0.11$ K. However, the difference in forcing agents between the two scenarios leads to a significant change in global precipitation that reaches $15 \pm 11$ mm yr$^{-1}$ in 2100. The change in precipitation is consistent with our results for the GeoMIP experiments and what we know of the global water cycle (Shine et al., 2015).

### 530    7.3. Variants focused on carbon-cycle (C4MIP)

     The C4MIP (Jones et al., 2016) experiments *ssp534-over-bgc* and *ssp585-bgc* differ from *ssp534-over* and *ssp585* in that the prescribed $CO_2$ does not affect the total radiative forcing , thus causing a lower change in GSAT and maintaining a relatively high carbon sinks efficiency. Figure 12 shows both carbon sinks under the variants and the base scenarios. Note that the *-bgc* experiments stem from a different historical simulation (*hist-bgc*). Under the high warming scenarios *ssp585*, climate change

reduces the oceanic carbon sink by $1.93 \pm 0.69$ PgC.yr$^{-1}$ and the net land carbon flux by $4.31 \pm 1.93$ PgC.yr$^{-1}$ in 2100. Under the overshoot scenario *ssp534-over*, this difference is lower, owing to its declining atmospheric $CO_2$. Removing the impact of climate change on the carbon cycle increases the land carbon stock by $269 \pm 52$ PgC in *ssp534-over*, but by $501 \pm 117$ PgC in *ssp585* in 2100, due to the higher warming in the latter case. We note that the permafrost carbon stock drives most of the changes, because if permafrost is ignored in the *bgc* variant, these changes are reduced to $57 \pm 32$ PgC and $131 \pm 77$ PgC in

*ssp534-over* and *ssp585* respectively.

### 7.4. Variants focused on land-use change (LUMIP & CDRMIP)

     LUMIP introduced variants of regular scenarios in which alternative land-use and land-cover change drivers coming from another scenario are prescribed (Lawrence et al., 2016), some of which being used in CDRMIP to assess afforestation (Keller et al., 2018). Two such experiments are the pessimistic *ssp585* and *ssp370* combined with the land-use activities of the

optimistic *ssp126* (named *ssp585-ssp126Lu* and *ssp370-ssp126Lu*, respectively). A third experiment consists in using the land-use of *ssp370* but under *ssp126*. (named *ssp126-ssp370Lu*). Comparisons of these experiments with their regular counterparts are shown in Figure 13. As expected, changing the land-use scenario roughly replaces one SSP's land-use emissions by





another's, albeit with some slight differences in the later stage of the simulations (i.e. after 2050) when atmospheric $CO_2$ and climate are significantly different from the reference scenario's, which has an impact in OSCAR because of transiently

changing land carbon densities. The effect on the land carbon sink is also quantified, showing that sink capacity can be preserved by conserving natural ecosystems, although it remains a relatively small effect in absolute value. We note that the ability of properly isolating both effects (on land-use emissions and on the sink) is a specific feature of OSCAR that stems from the formulation of its land carbon cycle (Gasser et al., 2020; Gasser and Ciais, 2013), and we do not expect many complex ESMs to be able to provide such a partitioning. The overall effect on land carbon stock change in 2100 is 48 ± 15 PgC, 76 ±

28 PgC and -65 ± 23 PgC, in the *ssp585-ssp126Lu*, *ssp370-ssp126Lu* and *ssp126-ssp370Lu* scenarios respectively. While the land carbon stocks are affected, the change in land cover also affects the planetary albedo. The radiative forcing from albedo of land cover change are exchanged between *ssp126* and *ssp370*, but changes remain below 0.1 W.m$^{-2}$. The net combined effect on projected temperature cannot be estimated, however, because these experiments are concentration-driven.

### 7.5.  Variant focused on SRM (GeoMIP)

In addition to the few idealized experiments of GeoMIP (Kravitz et al., 2015) that are accessible to OSCAR, one scenario variant focusing on SRM was also feasible. The *G6solar* experiment stems from *ssp585*, but the solar constant is changed from 2020 onwards to compensate the radiative forcing of *ssp585* and match the one of *ssp245*. As shown in Figure 14, although the GSAT of *G6solar* is brought to a level comparable to *ssp245*, some difference remains. The change in solar constant is calculated ex-ante as the difference from the radiative forcing of *ssp245* to *ssp585*, which by construction excludes some

feedbacks caused by this change and (as with *G1* and *G2*) does not fully cancel the change in global precipitation. Consequently, the carbon stocks still increase in *G6solar*, even more than in *ssp585* thanks to the lower GSAT and despite lower global precipitation.

### 7.6.  Differences between RCPs and SSPs (RCMIP)

Initially, the SSPs scenarios were designed to reach the RF of RCPs in 2100, to provide a common grid for reading and

comparing all the SSPs scenarios. Hence, the same four RF targets chosen in CMIP5 with the RCPs (2.6 W.m$^{-2}$, 4.5 W.m$^{-2}$, 6.0 W.m$^{-2}$, 8.5 W.m$^{-2}$) have also been chosen in CMIP6 with four out of the eight SSPs used. Yet, CMIP6 ESMs did not run RCPs, because these scenarios are not part of the CMIP6 experiments. Therefore, the difference between RCPs projections in CMIP5 and SSPs projections in CMIP6 under the same RF targets are due to both a change in the generation of ESMs and a change in scenarios. In Figure 15, we represent both RCPs and SSPs under the same version of OSCAR, showing the difference

due to the sole change in scenarios. These scenarios use different drivers, as illustrated with the atmospheric $CO_2$ prescribed to these concentration-driven experiments, usually with higher $CO_2$ concentrations in the CMIP6 version. Except for the 8.5 target, the RF tends to be also higher in the CMIP6 version, compared to the CMIP5 version, meaning changes in other drivers are not enough to balance the $CO_2$ increase. While the 2.6 W.m$^{-2}$ and 8.5 W.m$^{-2}$ targets are reached in 2100, the 4.5 W.m$^{-2}$ and 6.0 W.m$^{-2}$ are not. However, our results can be compared to those of MAGICC in these two cases (Iiasa, 2018b), and both



reduced-complexity models are consistent. Because of the similar RF targets, GSAT are relatively similar over the 21st century, but RCPs and SSPs tend to dissociate later on. In 2300, moving from RCPs to SSPs changes GSAT by $18 \pm 8\%$, $9 \pm 3\%$, $5 \pm 2\%$ and $-6 \pm 1\%$ in the four tested scenarios, respectively. Differences in other key variables such as the carbon sinks logically respond to these differences in atmospheric $CO_2$ and global temperature change, as also shown in Figure 15.

## 8.  Concluding remarks

In this paper, we have presented the setup used with OSCAR v3.1 to run 75 CMIP6 and 24 additional experiments from RCMIP. We have used the primary results of these simulations to discuss the overall behaviour and performance of our model, comparing our results to those of state-of-the-art complex models whenever possible. Follows a brief summary of the model's main limitations.

First, the model tends to be unstable under high warming scenarios. This comes mostly from the ocean carbon cycle module
whose stability is not ensured under our chosen differential system solving scheme, which is also worsened by the stratification feedback that was introduced in v2.2 (Gasser et al., 2017). This pleads for a revamp of this module. Second, despite a clear improvement of the land carbon cycle module in v3.1 (Gasser et al., 2020), its unconstrained transient response remains wider than the ranges from CMIP5 or CMIP6, which makes the constraining step a strong requirement of any simulation with OSCAR. In its current state, however, the constraining step appears to favor parameterizations with a strong $CO_2$-fertilization
effect. The land carbon cycle also exhibits a sensitivity to climate change that is too low compared to complex models, mostly those without permafrost, thus calling for an improved calibration. A potential track would be to account for correlations between parameters. Third, the constrained climate module shows a relatively low ECS and a rather narrow uncertainty range. Introducing extra parameters for the heat uptake feedback (Geoffroy et al., 2013a) and possibly non-linear Charney feedbacks (Bloch-Johnson et al., 2015) would likely help in a more flexible constraining approach. This third point is the reason behind
most of the difference between OSCAR and CMIP6 temperature projections shown in Table 5. Fourth, although most of the non-$CO_2$ species are reasonably simulated, the effects of tropospheric ozone and total aerosols tend to be overestimated, and the whole aerosol module behaves rather linearly. OSCAR would indeed benefit from further work on short-lived species, although this could prove a challenging endeavour given the aggregated formulation of the model and the uncertainties. Finally, we have illustrated how observational constraints can be used to inform projections, how it may affect the results, such as the
strong decrease of uncertainties in projections. Given the growing importance of these constraints (Tokarska et al., 2020; Nicholls et al., 2021), this calls for investigating computationally efficient and physically sensible ways of doing so with OSCAR.

In spite of those limitations, we have demonstrated that OSCAR behaves as one should expect from an Earth system model. Applying our two post-processing steps (exclusion and constraining) somewhat overcomes the model's limitations, and the
resulting quantitative behaviour of OSCAR remains largely satisfactory. In some cases, we have also shown that OSCAR differs from complex models, due to the features of OSCAR that are not yet found in most complex models, such as fully





interactive atmospheric chemistry that would allow $CH_4$ and $N_2O$ to be emission-driven, and endogenous simulation of $CH_4$ emissions from wetlands, $CO_2$ and $CH_4$ permafrost and biomass burning. Therefore, some of the results presented here have scientific interests that go beyond the pure model evaluation perspective. These valuable insights for other projects will be

presented in separate papers, but all outputs from the simulations presented here are already publicly available (and more can be requested from the authors). Finally, this study will be the basis for a more systematic assessment of the model's performance, as we will use the standardised CMIP6 and RCMIP simulations to diagnose future versions of OSCAR and to compare them with older versions. This will provide the wider community with a benchmark of the model, hopefully spreading interest in this open-source compact Earth system model.






**Acknowledgements**

TG and YQ acknowledge support from the European Research Council Synergy project "Imbalance-P" (grant ERC-2013-SyG-610028). This work is also part of the European Union's Horizon 2020 research and innovation project "CONSTRAIN" (grant #820829). This work was also partially funded by the Austrian Science Fund FWF project "ERM" (grant P31796-N29). The simulations were performed on the *ebro* cluster at IIASA. We also thank Zebedee Nicholls for providing us with the timeseries of global surface air temperature of CMIP6.


**Author contribution**

TG developed OSCAR. YQ processed the input data and set up the simulations. YQ and TG decided on the post-processing (exclusion and constraining). YQ executed the post-processing. YQ produced all the figures. YQ and TG drafted the manuscript. All authors contributed to the final analysis and to the manuscript.


**Competing interests**

The authors declare that they have no conflict of interest.

**Code and data availability**

The source code of OSCAR is available at https://github.com/tgasser/OSCAR. Results for some of the experiments performed are available on the repository of RCMIP phase 2 (https://doi.org/10.5281/zenodo.4269711). Additional data and scripts are available upon request to the corresponding author.



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



**Table 1. List of CMIP6 and RCMIP simulations run with OSCAR.** Standard names are used, and full description of the experiments are provided in references. Every experiment that is a scenario has been run with its extension up to 2500. A spin-up of 1000 years is associated to each of the 8 control experiments.

| MIP | Simulations |
| --- | --- |
| DECK (Eyring et al., 2016) | *1pctCO2, abrupt-4xCO2, esm-hist, historical, piControl, esm-piControl* |
| AerChemMIP (Collins et al., 2017) | *hist-1950HC, hist-piAer, hist-piNTCF, ssp370-lowNTCF* |
| C4MIP (Jones et al., 2016) | *1pctCO2-bgc, 1pctCO2-rad, esm-ssp585, hist-bgc, ssp534-over-bgc, ssp585-bgc* |
| CDRMIP (Keller et al., 2018) | *1pctCO2-CDR, esm-pi-cdr-pulse, esm-pi-CO2pulse, esm-yr2010CO2-cdr-pulse, esm-yr2010CO2-CO2pulse, esm-yr2010CO2-control, esm-yr2010CO2-noemit, esm-ssp534-over, esm-ssp585-ssp126Lu, yr2010CO2* |
| DAMIP (Gillett et al., 2016) | *hist-aer, hist-CO2, hist-GHG, hist-nat, hist-sol, hist-stratO3, hist-volc, ssp245-aer, ssp245-CO2, ssp245-GHG, ssp245-nat, ssp245-sol, ssp245-stratO3, ssp245-volc* |
| LUMIP (Lawrence et al., 2016) | *esm-ssp585-ssp126Lu, hist-noLu, land-cClim, land-cCO2, land-crop-grass, land-hist, land-hist-altLu1, land-hist-altLu2, land-hist-altStartYear, land-noLu, land-noShiftCultivate, land-noWoodHarv, ssp126-ssp370Lu, ssp370-ssp126Lu, land-piControl, land-piControl-altLu1, land-piControl-altLu2, land-piControl-altStartYear* |
| GeoMIP (Kravitz et al., 2015) | *G1, G2, G6solar* |
| ScenarioMIP (O'neill et al., 2016) | *ssp119, ssp126, ssp245, ssp370, ssp434, ssp460, ssp534-over, ssp585* |
| ZECMIP (Jones et al., 2019) | *esm-1pctCO2, esm-1pct-brch-750PgC, esm-1pct-brch-1000PgC, esm-1pct-brch-2000PgC, esm-bell-750PgC, esm-bell-1000PgC, esm-bell-2000PgC* |
| RCMIP (Nicholls et al., 2020) | *1pctCO2-4xext, abrupt-0p5xCO2, abrupt-2xCO2, esm-abrupt-4xCO2, esm-histcmip5, esm-rcp26, esm-rcp45, esm-rcp60, esm-rcp85, esm-ssp119, esm-ssp126, esm-ssp245, esm-ssp370, esm-ssp370-lowNTCF, esm-ssp434, esm-ssp460, historical-CMIP5, rcp26, rcp45, rcp60, rcp85, ssp585-ssp126Lu, esm-piControl-CMIP5, piControl-CMIP5* |






**Table 2: Metrics of the climate system (ECS, TCR and TCRE).** Metrics are provided for OSCAR v3.1 constrained using observations, and unconstrained. Values are provided as mean ± standard deviation, median and the [5%-95%] confidence interval. As explained in section 4.1, the ECS in OSCAR may be calculated using its parameters, or simply as the temperature at the end of *abrupt-2xCO2*. These values are compared to the ECS of (Meehl et al., 2020). The same source provides the values for the TCR. The TCRE of CMIP5 is compared to (Gillett et al., 2013). Values from RCMIP phase 2 (Nicholls et al., 2021) come from different souces: (Sherwood et al., 2020) for the ECS, (Tokarska et al., 2020) for the TCR and (Arora et al., 2020) for the TCRE.

| | | OSCAR v3.1 | | CMIP5 | CMIP6 | RCMIP, phase 2 |
| --- | --- | --- | --- | --- | --- | --- |
| | | Unconstrained | Constrained | | | |
| ECS (K) | Parameter value | 3.17 ± 0.63 3.28 [2.36-4.25] | 2.78 ± 0.47 2.63 [2.36-3.75] | 3.2 ± 0.7 | 3.7 ± 1.1 | 3.10 [2.30-4.70] |
| | End of *abrupt-2xCO2* | 2.74 ± 0.52 2.61 [2.02-3.67] | 2.52 ± 0.33 2.45 [2.08-3.22] | | | |
| TCR (K) | | 1.78 ± 0.28 1.77 [1.37-2.26] | 1.66 ± 0.16 1.62 [1.41-1.96] | 1.8 ± 0.40 | 2.0 ± 0.4 | 1.64 [0.98-2.29] |
| TCRE (K EgC⁻¹) | | 1.67 ± 0.40 1.63 [1.08-2.37] | 1.44 ± 0.20 1.41 [1.15-1.82] | 1.63 ± 0.48 [0.8-2.4] | 1.77 ± 0.37 | 1.77 [1.03-2.51] |





**Table 3: Metrics of the carbon-cycle (β and $\gamma$) from the C4MIP experiments.** Metrics are provided for OSCAR v3.1 constrained using observations, and unconstrained. As explained by (Arora et al., 2013b), different values for the metrics are calculated depending on the combination of experiments used: R stands for radiative (*1pctCO2-rad*), B for biogeochemical (*1pctCO2-bgc*) and F for full (*1pctCO2*). The change in the land carbon stocks includes permafrost carbon. Results from CMIP5 and CMIP6 are provided by C4MIP (Arora et al., 2020).

| Time | Model | Method $\beta$ | $\beta$ (PgC ppm$^{-1}$) | | Method $\gamma$ | $\gamma$ (PgC K$^{-1}$) | |
|------|-------|---------|------|-------|---------|------|-------|
| | | | Land | Ocean | | Land | Ocean |
| 2xCO2 | OSCAR v3.1 constrained | R-B, B-F | 1.26 ± 0.47 | 1.05 ± 0.03 | R-B, R-F | -34.7 ± 18.9 | -13.0 ± 0.7 |
| | | R-F | 1.21 ± 0.44 | 1.00 ± 0.06 | B-F | -43.2 ± 23.8 | -21.6 ± 6.3 |
| | OSCAR v3.1 unconstrained | R-B, B-F | 1.14 ± 0.64 | 1.05 ± 0.03 | R-B, R-F | -30.8 ± 20.5 | -13.0 ± 0.7 |
| | | R-F | 1.10 ± 0.61 | 1.00 ± 0.05 | B-F | -37.6 ± 26.4 | -21.0 ± 5.7 |
| | CMIP5 | B-F | 1.15 ± 0.63 | 0.95 ± 0.07 | B-F | -37.0 ± 25.5 | -9.4 ± 2.7 |
| | CMIP6 | B-F | 1.22 ± 0.40 | 0.91 ± 0.09 | B-F | -34.1 ± 38.4 | -8.6 ± 2.9 |
| 4xCO2 | OSCAR v3.1 constrained | R-B, B-F | 1.06 ± 0.41 | 0.94 ± 0.03 | R-B, R-F | -47.7 ± 23.8 | -17.7 ± 1.3 |
| | | R-F | 0.95 ± 0.37 | 0.86 ± 0.08 | B-F | -72.3 ± 37.4 | -37.1 ± 13.6 |
| | OSCAR v3.1 unconstrained | R-B, B-F | 0.96 ± 0.57 | 0.94 ± 0.03 | R-B, R-F | -43.3 ± 25.5 | -17.7 ± 1.3 |
| | | R-F | 0.87 ± 0.50 | 0.86 ± 0.07 | B-F | -63.1 ± 41.5 | -35.5 ± 12.4 |
| | CMIP5 | B-F | 0.93 ± 0.49 | 0.82 ± 0.07 | B-F | -57.9 ± 38.2 | -17.3 ± 3.8 |
| | CMIP6 | B-F | 0.97 ± 0.40 | 0.78 ± 0.07 | B-F | -45.1 ± 50.6 | -17.2 ± 4.9 |






**Table 4: Zero Emissions Commitments at 25, 50, 90 and 500 years after emission cease.** Only the ZEC for the experiment *esm-1pct-brch-1000PgC* are shown here, for comparison to results of ZECMIP. The full evolution of this experiment is shown in Figure 8.

|  | $ZEC_{25}$ (K) | $ZEC_{50}$ (K) | $ZEC_{90}$ (K) | $ZEC_{500}$ (K) |
|---|---|---|---|---|
| OSCAR v3.1 | $-0.01 \pm 0.07$ | $-0.02 \pm 0.09$ | $-0.01 \pm 0.11$ | $-0.21 \pm 0.13$ |
| ZECMIP (Macdougall et al., 2020) | $-0.01 \pm 0.15$ | $-0.06 \pm 0.19$ | $-0.11 \pm 0.23$ |  |






**Table 5: Attribution of historical and future climate change.** These contributions come either from experiments in which only the concerned forcing was prescribed (DAMIP), or from experiments in which it was removed (other MIPs). In either cases, non-linearities are ignored.

| Experiments | | | GSAT w.r.t. 1850-1900 (K) | | RF (W.m$^{-2}$) | |
|---|---|---|---|---|---|---|
| | 2006-2015 | 2091-2100 | 2006-2015 | 2091-2100 | 2006-2015 | 2091-2100 |
| All forcings | *historical* | *ssp245* | $0.98 \pm 0.19$ | $2.53 \pm 0.25$ | $2.07 \pm 0.42$ | $4.62 \pm 0.29$ |
| WMGHGs[†] | *hist-GHG* | *ssp245-GHG* | $1.24 \pm 0.12$ | $2.67 \pm 0.29$ | $2.53 \pm 0.13$ | $4.73 \pm 0.27$ |
| NTCFs[‡] | *hist-aer* | *ssp245-aer* | $-0.26 \pm 0.22$ | $-0.15 \pm 0.12$ | $-0.48 \pm 0.36$ | $-0.16 \pm 0.12$ |
| *id.* | *historical - hist-piNTCF* | -- | $-0.25 \pm 0.21$ | -- | $-0.46 \pm 0.35$ | -- |
| Natural forcings | *hist-nat* | *ssp245-nat* | $\sim 0.03$ | $\sim 0.01$ | $\sim 0.09$ | $\sim 0.00$ |
| CO$_2$ | *hist-CO2* | *ssp245-CO2* | $0.74 \pm 0.07$ | $2.03 \pm 0.22$ | $1.52 \pm 0.09$ | $3.70 \pm 0.24$ |
| CO$_2$ radiative effect only | *historical - hist-bgc* | -- | $0.75 \pm 0.08$ | -- | $1.55 \pm 0.04$ | -- |
| CFCs and HCFCs[†] | *historical - hist-1950HC* | -- | $0.13 \pm 0.02$ | -- | $0.27 \pm 0.03$ | -- |
| Stratospheric O$_3$ | *hist-stratO3* | *ssp245-stratO3* | $-0.03 \pm 0.03$ | $-0.02 \pm 0.03$ | $-0.07 \pm 0.06$ | $-0.02 \pm 0.05$ |
| Aerosols | *historical - hist-piAer* | -- | $-0.33 \pm 0.20$ | -- | $-0.63 \pm 0.33$ | -- |
| Solar activity | *hist-sol* | *ssp245-sol* | $\sim 0.02$ | $\sim 0.01$ | $\sim 0.03$ | $\sim 0.02$ |
| Volcanic activity | *hist-volc* | *ssp245-volc* | $\sim 0.01$ | $\sim -0.01$ | $\sim 0.06$ | $\sim -0.02$ |
| Land-use change | *historical - hist-noLu* | -- | $-0.03 \pm 0.03$ | -- | $-0.05 \pm 0.05$ | -- |

[†] In these experiments, because the atmospheric concentration of WMGHGs is prescribed, the indirect effects on tropospheric O$_3$ (from CH$_4$), stratospheric H$_2$O (from CH$_4$) and stratospheric O$_3$ (from N$_2$O and halogenated compounds) are also included.

[‡] The effects listed in the previous note on WMGHGs are excluded from this experiment. Tropospheric O$_3$ does vary, however, but only because of the emission of ozone precursors and not because of varying atmospheric CH$_4$. Black carbon deposition on snow is also included in this experiment.





**Table 6: Projected atmospheric CO₂, RF and GSAT in SSPs.** Concentration- and emission-driven experiments are shown and compared to available CMIP6 projections. Values in bold are assumptions or inputs. Experiments whose name start with *esm-* are emission-driven; others are concentration-driven. GSAT from CMIP6 are provided as mean and standard deviation as well, with the number of models available in parenthesis. Here, projections from OSCAR are constrained to observations, while CMIP6 results are raw, without any constraints (Tokarska et al., 2020).

| experiments | models | ERF (W m⁻²) | GSAT w.r.t. 1850-1900 (K) | | | | CO₂ (ppm) | |
|---|---|---|---|---|---|---|---|---|
| | | 2100 | 2041-2050 | 2091-2100 | 2291-2300 | 2491-2500 | 2100 | 2300 |
| *esm-ssp585* | OSCAR | 8.40 ± 0.57 | 2.02 ± 0.22 | 3.99 ± 0.40 | 6.31 ± 0.83 | 6.29 ± 0.88 | 1058 ±63 | 1729 ± 148 |
| *esm-ssp585* | CMIP6 | | 2.41 ± 1.67 (3) | 5.14 ± 3.92 (2) | | | | |
| *ssp585* | OSCAR | 8.76 ± 0.50 | 2.04 ± 0.19 | 4.16 ± 0.38 | 7.05 ± 0.87 | 7.24 ± 0.93 | 1135 | 2162 |
| *ssp585* | CMIP6 | | 2.72 ± 1.51 (17) | 6.19 ± 3.13 (17) | 13.51 ± 5.87 (2) | | **1135** | **2162** |
| *esm-ssp370* | OSCAR | 7.04 ± 0.66 | 1.85 ± 0.25 | 3.32 ± 0.35 | 5.54 ± 0.74 | 5.56 ± 0.80 | 809 ± 47 | 1200 ± 109 |
| *ssp370* | OSCAR | 7.41 ± 0.58 | 1.87 ± 0.21 | 3.50 ± 0.32 | 6.24 ± 0.75 | 6.41 ± 0.81 | 867 | 1483 |
| *ssp370* | CMIP6 | | 2.51 ± 1.48 (18) | 5.1 ± 2.84 (16) | | | **867** | **1483** |
| *esm-ssp460* | OSCAR | 5.32 ± 0.50 | 1.80 ± 0.23 | 2.68 ± 0.30 | 3.43 ± 0.51 | 3.34 ± 0.55 | 629 ± 35 | 667 ± 49 |
| *ssp460* | OSCAR | 5.64 ± 0.40 | 1.82 ± 0.19 | 2.84 ± 0.27 | 3.91 ± 0.47 | 3.89 ± 0.50 | 668 | 769 |
| *ssp460* | CMIP6 | | 2.46 ± 1.28 (4) | 4.24 ± 1.80 (4) | | | **668** | **769** |
| *esm-ssp245* | OSCAR | 4.63 ± 0.43 | 1.72 ± 0.21 | 2.38 ± 0.28 | 2.59 ± 0.41 | 2.40 ± 0.42 | 578 ± 31 | 565 ± 35 |
| *ssp245* | OSCAR | 4.86 ± 0.31 | 1.75 ± 0.17 | 2.50 ± 0.25 | 2.92 ± 0.37 | 2.79 ± 0.37 | 603 | 621 |
| *ssp245* | CMIP6 | | 2.41 ± 1.33 (15) | 3.63 ± 1.82 (15) | | | **603** | **621** |
| *esm-ssp534-over* | OSCAR | 2.93 ± 0.37 | 2.00 ± 0.22 | 1.73 ± 0.25 | 1.16 ± 0.23 | 1.02 ± 0.23 | 458 ± 23 | 374 ± 12 |
| *ssp534-over* | OSCAR | 3.36 ± 0.27 | 2.04 ± 0.19 | 1.95 ± 0.22 | 1.40 ± 0.20 | 1.29 ± 0.19 | 497 | 398 |
| *ssp534-over* | CMIP6 | | 2.88 ± 0.84 (6) | 3.08 ± 1.06 (6) | 1.85 ± 0.66 (2) | | **497** | **398** |
| *esm-ssp434* | OSCAR | 3.45 ± 0.40 | 1.64 ± 0.20 | 1.87 ± 0.24 | 1.51 ± 0.28 | 1.44 ± 0.29 | 451 ± 21 | 371 ± 15 |
| *ssp434* | OSCAR | 3.70 ± 0.31 | 1.65 ± 0.17 | 2.00 ± 0.21 | 1.73 ± 0.24 | 1.68 ± 0.25 | 473 | 392 |
| *ssp434* | CMIP6 | | 2.36 ± 1.1 (5) | 3.23 ± 1.32 (5) | | | **473** | **392** |
| *esm-ssp126* | OSCAR | 2.66 ± 0.29 | 1.54 ± 0.18 | 1.49 ± 0.21 | 1.17 ± 0.20 | 1.02 ± 0.20 | 439 ± 18 | 381 ± 11 |
| *ssp126* | OSCAR | 2.80 ± 0.20 | 1.58 ± 0.15 | 1.58 ± 0.17 | 1.31 ± 0.18 | 1.21 ± 0.18 | 446 | 396 |
| *ssp126* | CMIP6 | | 2.21 ± 1.1 (17) | 2.38 ± 1.17 (17) | 1.68 ± 0.7 (2) | | **446** | **396** |
| *esm-ssp119* | OSCAR | 2.0 ± 0.25 | 1.39 ± 0.17 | 1.15 ± 0.17 | 0.71 ± 0.15 | 0.61 ± 0.15 | 383 ± 12 | 334 ± 6 |
| *ssp119* | OSCAR | 2.14 ± 0.18 | 1.44 ± 0.14 | 1.24 ± 0.15 | 0.82 ± 0.13 | 0.74 ± 0.13 | 394 | 342 |
| *ssp119* | CMIP6 | | 2.36 ± 1.07 (6) | 2.12 ± 0.92 (2) | | | **394** | **342** |




**Figure 1: Effect of the constraining step.** The histograms are the results of OSCAR v3.1, with plain lines being for the constrained version, while the dotted lines are for the unconstrained version. Horizontal lines correspond to the average plus or minus one standard deviation. Cumulative compatible emissions from *historical-CMIP5* are calculated over 1850-2011, while those of the RCPs are calculated over 2012-2100.





**Figure** 2: **Emission- and concentration-driven historical scenarios.** The plain lines are the averages, and the shaded areas are the ±
1 standard deviation ranges. The fossil-fuel CO2 emissions for the concentrations-driven historical are the compatible emissions, whereas
those for the emissions-driven esm-hist are directly prescribed to OSCAR. Radiative forcings under esm-hist are not represented, for they
are too close from the concentrations-driven historical. Radiative forcings are with respect to 1750. The sources for the observations are
(Friedlingstein et al., 2020) for GCB2020, (Hartmann et al., 2013) for the 'AR5 WG1 Ch2', (Ciais et al., 2013b) for 'AR5 WG1 Ch3' and
(Myhre et al., 2013) for 'AR5 WG1 Ch8'. The 90% ranges provided by AR5 are converted to ± 1 standard deviation ranges.




**Figure 3: Abrupt idealized experiments.** In the left panel, the plain lines represent the average change in surface air temperature, and its ± 1 standard deviation ranges using shaded areas. The three middle panels show the contributions to the total RF at equilibrium. Individual contributions from stratospheric $O_3$ and deposition of BC on snow are inferior to 0.1 W.m$^{-2}$ in the *abrupt-4xCO2*, and have not been represented for clarity. The three right panels are the distributions of the ECS, calculated using equilibrium temperature, and thus including all the feedbacks of OSCAR. The horizontal plain line is the ECS average and ± 1 standard deviation range. These values with Pearson's moment coefficient of skewness are provided in the legend.




**Figure 4: Experiments with 1% increase in the atmospheric CO₂.** The plain lines are the averages, and the shaded areas are the ± 1 standard deviation ranges.




**Figure 5: Experiments from GeoMIP compared to their DECK counterpart.** The plain lines are the averages, and the shaded areas are the ± 1 standard deviation ranges.




**Figure 6: Reversibility experiment from CDRMIP.** The orange lines correspond to the ramp-up of *1pctCO2-cdr*, the blue line to its ramp-down and the grey line to the 1000 years with constant atmospheric CO₂. The plain lines are the averages, and the shaded areas are the ± 1 standard deviation ranges.






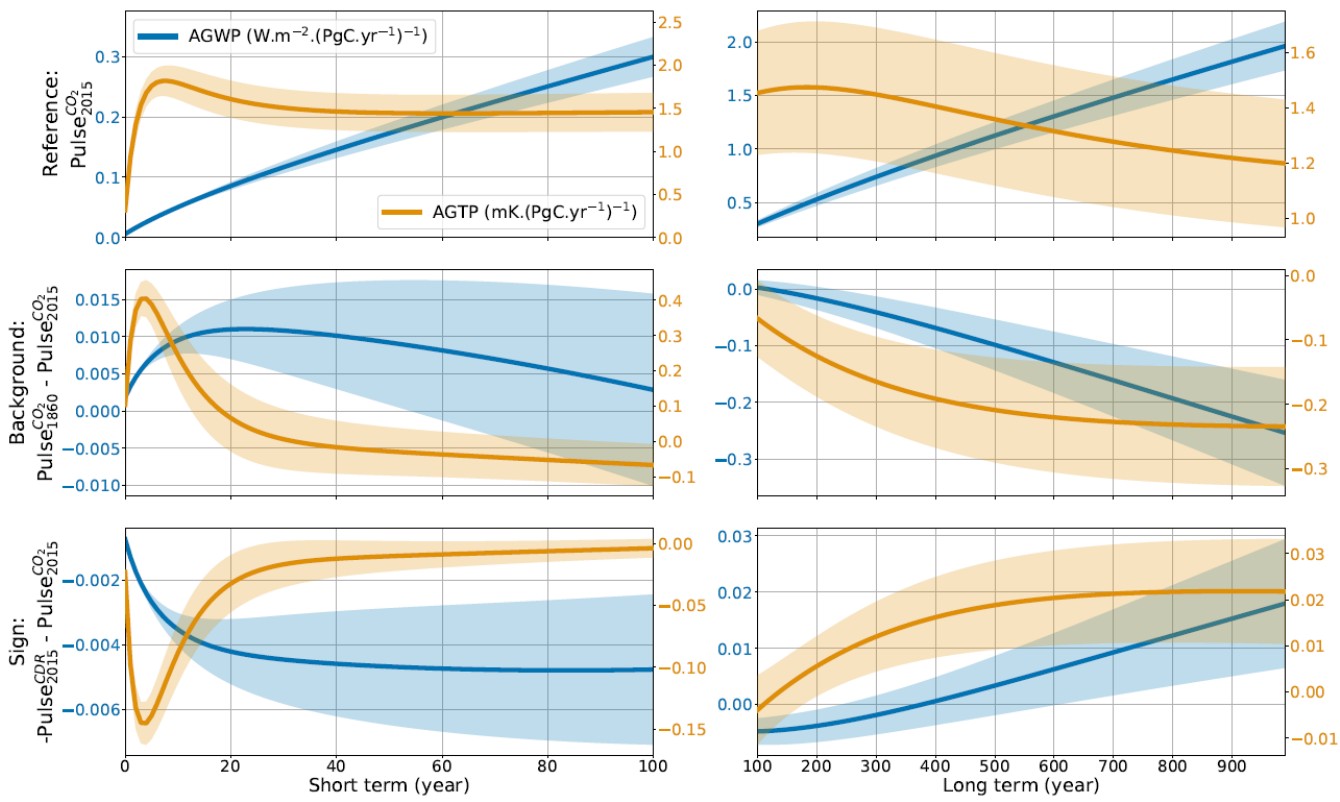

**Figure 7: AGWP (blue) and AGTP (orange) of CO₂ for 100PgC of CO₂ emissions under actual environmental conditions.** The dependency of this reference to a change of background is on the second line. The dependency to the sign of the pulse, emissions or removal, is on the third line. The lines are the averages, and the shaded areas are the ± 1 standard deviation ranges.


**Figure 8: Change in global mean surface temperature for branched experiments (top panels) and bells experiments (bottom panels).** The results over the zero-emission phase are shifted along the time axis so that $t = 0$ corresponds to the time of cessation of emission. The lines are the averages, and the shaded areas are the $\pm 1$ standard deviation ranges.




**Figure 9: Land-use experiments from LUMIP.** The first row of the figure corresponds to the reference experiment (*land-hist*) while other rows show sensitivity experiments as a difference to *land-hist*. *land-hist*. *land-hist-altStartYear* is shown only from 1850 despite starting in 1700. The lines are the averages, and the shaded areas are the ± 1 standard deviation ranges.






**Figure 10: Global projections following the main CMIP6 scenarios in concentration-driven mode.** Extensions are shown only up to 2300. The lines are the averages, and the shaded areas are the ± 1 standard deviation ranges.



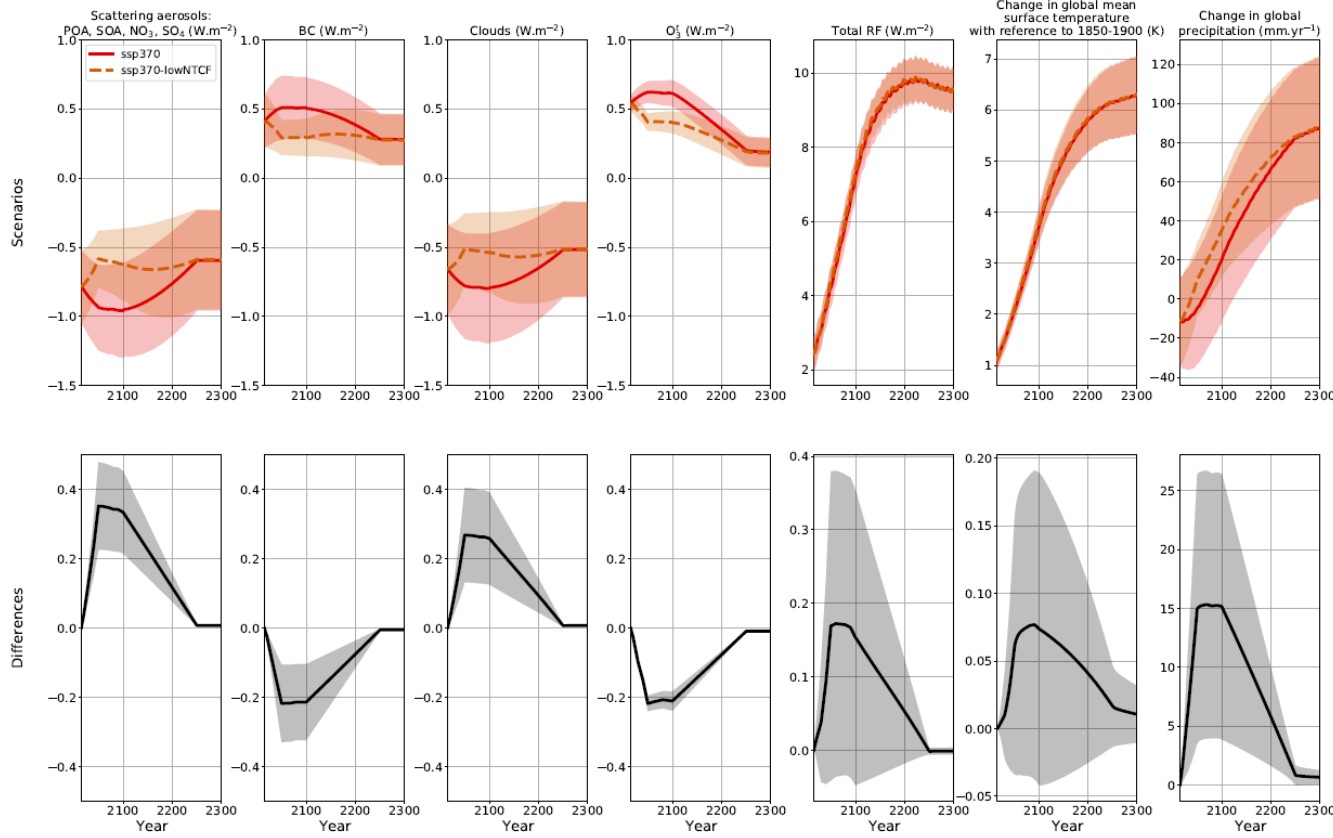

**Figure 11: Effect of lower NTCF emissions in the SSP3-7.0.** Extensions are shown only up to 2300. The lines are the averages, and
the shaded areas are the ± 1 standard deviation ranges.



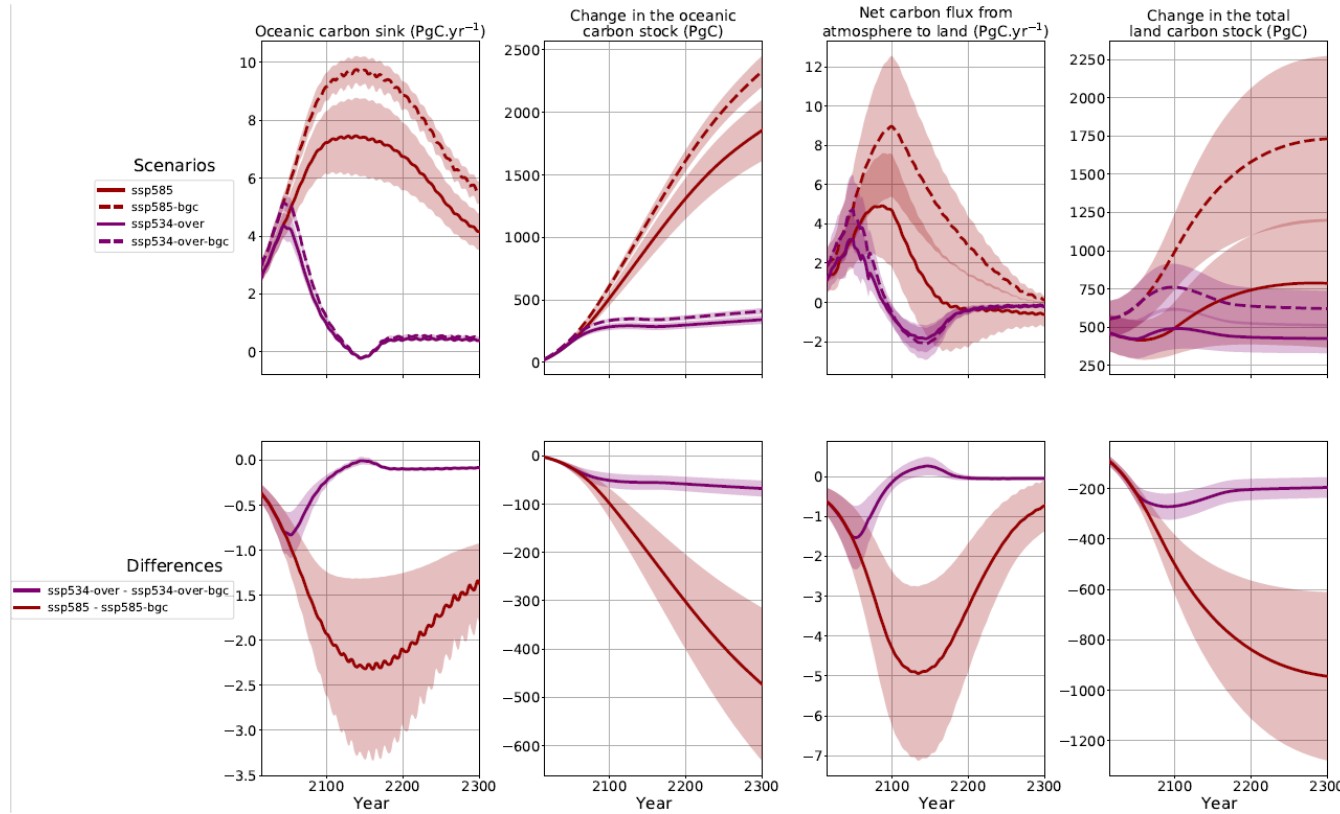

**Figure 12: Effect of climate change on the carbon cycle in the scenarios *ssp534-over* and *ssp585*.** The net flux from atmosphere from land is the sum of the land carbon sink, $CO_2$ emissions from land-use and land-cover change, and $CO_2$ and $CH_4$ emissions from permafrost. The changes in the total land carbon stock include those in the permafrost. Note that the increased uncertainty in the ocean sink before 2250 is an artefact of our exclusion procedure (see text on post-processing) that cannot capture some Monte Carlo members that already started diverging. Extensions are shown only up to 2300. The lines are the averages, and the shaded areas are the $\pm 1$ standard deviation ranges.

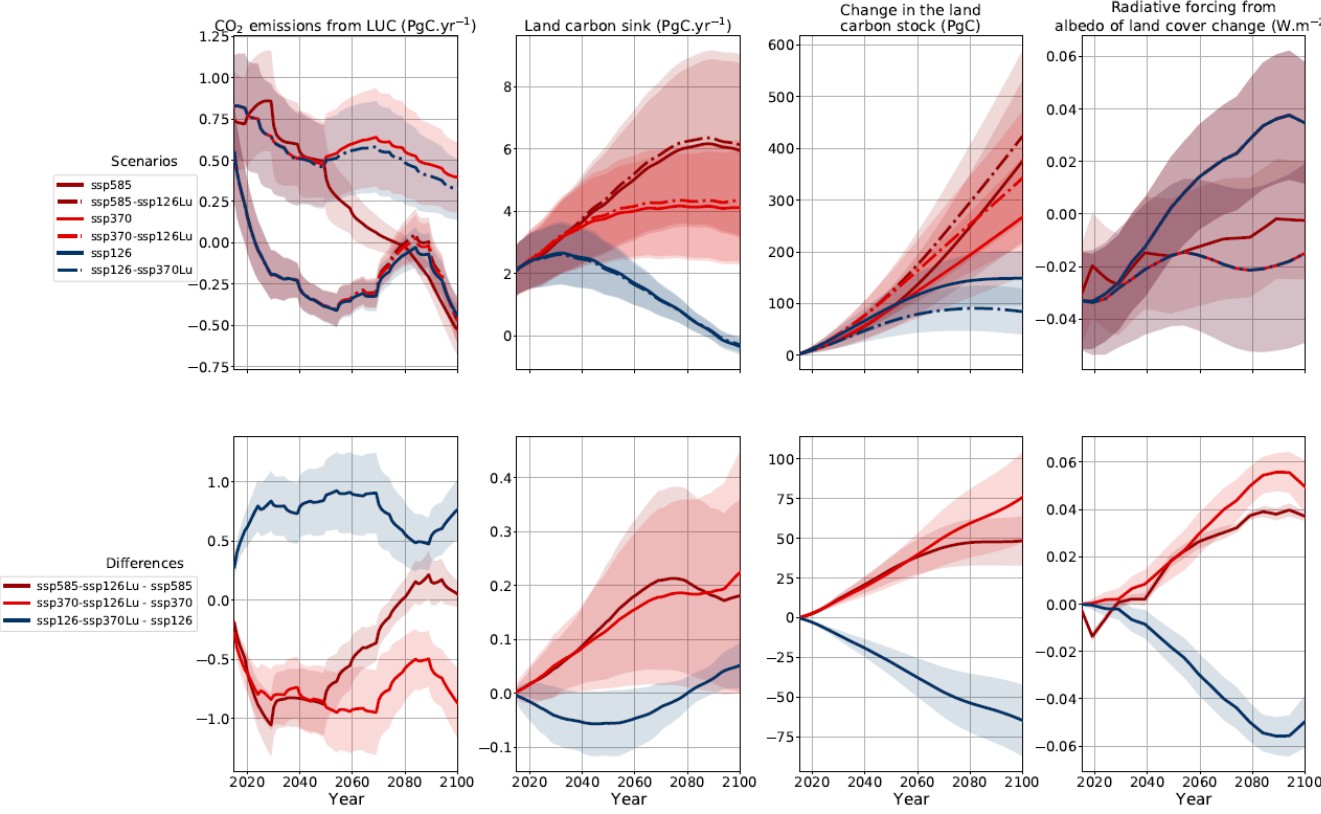

**Figure 13: Effect of alternative land-use and land-cover change drivers in the scenarios *ssp126*, *ssp370* and *ssp585*.** Here, the changes in the land carbon stock does not include the changes in the permafrost. The lines are the averages, and the shaded areas are the ± 1 standard deviation ranges.



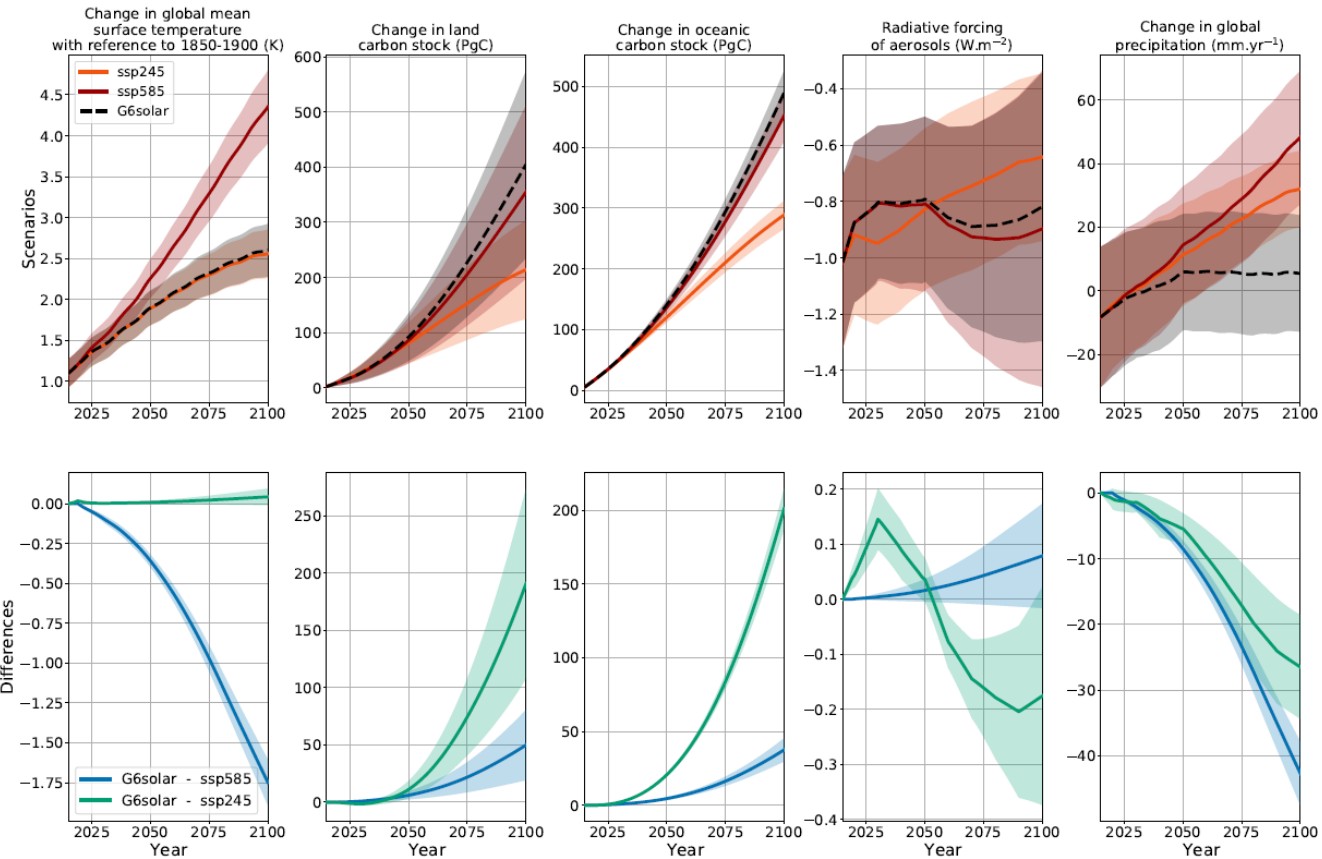

**Figure 14: Effect of introducing SRM in the SSP5-8.5 to reach the SSP2-4.5.** The lines are the averages, and the shaded areas are the ± 1 standard deviation ranges.



1170

**Figure 15: Comparison between RCPs (CMIP5) and SSPs (CMIP6).** The lines are the averages, and the shaded areas are the ± 1 standard deviation ranges.