# Peer review of "CMIP6 simulations with the compact Earth system model OSCAR v3.1"

_Geoscientific Model Development, 2021_

## Referee Comment (RC1)

**General comments**

Quilcaille et al. present a summary of a number of CMIP6-style simulations performed with the reduced complexity earth system model OSCARv3.1. They briefly describe the model, how they constrained it to create probabilistic results and then go through the results of their numerous experiments.

The paper is clearly the result of a massive amount of effort, and having this description of OSCAR in the literature is beneficial. My issue with the current presentation is that it lacks any punch. There are so many results, and comparatively so little discussion, that it's hard to know what I'm meant to take away from this apart from, "We ran our model heaps". That's not to say that there aren't really interesting results in the paper, it's just that they're overly hard to find.

I think the paper would benefit greatly from improved focus. Many of the experiments require extensive discussion and further exploration. There doesn't appear to be space for all of them in this paper. For those that cannot be fully explored, I think it is better to save them for future papers where they can be explored appropriately rather than having partial explorations (also because I don't think partial explorations belong in the scientific literature).

**Major concerns**

**Lack of focus**

As discussed above, the paper lacks focus. Results are presented from experiment after experiment, with no space to actually explore their implications or what to make of them. Obvious examples are Sections 7.2, 7.3 and 7.5. However, many of the sections left me wondering, "What is the point?"

This is unfortunate, because many of the results are very interesting. For example, it is surprising that the ZEC is much higher for 2000 PgC experiments but overall warming doesn't have the same non-linearity. However, there is no further exploration of this. Similarly, "the carbon stocks still increase in G6solar, even more than in ssp585 thanks to the lower GSAT and despite lower global precipitation." Is this what we would expect? Or does this point to a clear limitation of the model if we have less rain but somehow more carbon stocks? Eight different spin ups are done: how do they compare? What does this tell us about the way we make climate projections and any potential bias in the CMIP-style of doing things. The authors also write, "Our results cannot be compared to the final CDRMIP results yet, for they are unpublished, but they are consistent with those obtained with a model of intermediate complexity (Zickfeld et al., 2021)." However, they have a great tool to evaluate the questions: if they know how much to trust their model (which they should at the end of this evaluation), then they don't need to wait for the CDRMIP results and could write a great (separate) paper on their results now.

I would recommend the authors reconsider which results to present in this paper,

which belong in their own paper and which are best left out. This would probably significantly reduce the length of the paper. It would also improve the abstract, which is currently too long and contains too much detail (it could be re-written to just focus on key points: Emulators are needed, emulators need to be validated, here we examine OSCAR, strengths are X, weaknesses are Y, last sentence could stay as is).

**Writing**

The writing is very slow, i.e. it doesn't always make clear what the point is. I think this is partly due to a lack of focus as discussed above. It's partly due to being repetitive (the point about the need for validation is mentioned three times in the first paragraph). However, I think it is also partly due to phrasing. It might help to swap phrases like, "As illustrated in Table 4, OSCAR v3.1 estimates a ZEC (in the reference case of the esm-1pct-brch-1000PgC experiment) that is within the range of ZECMIP (Macdougall et al., 2020), although the long-term decrease seems to happen later in OSCAR." with more active formulations (that move all the table and figure references to parentheses) like "OSCAR v3.1 estimates a ZEC (in the reference case of the esm-1pct-brch-1000PgC experiment) that is within the range of ZECMIP (Macdougall et al., 2020), although the long-term decrease seems to happen later in OSCAR (Table 4)." (Yes, I acknowledge the irony of giving advice about punchy writing when my review is probably slightly rambling.)

The paper is also in need of a proofread. Reading it this time was overly difficult due to typing and other phrasing errors. This will take some effort, but it will greatly improve the experience for the reader.

**Vague claims of goodness**

The paper has some very vague claims of goodness. Two examples, "It reproduces the responses of complex ESMs, for all aspects of the Earth system.", and, "the resulting quantitative behaviour of OSCAR remains largely satisfactory". Both these claims are vague and subjective. I would simply remove them and all others like them from the paper, the reader can judge the quality for themselves based on the results (and likely, the 'good enough' level will change depending on the application of interest).

**Specific comments**

1. "Similar reasons explain why a pulse of carbon removal cools the atmosphere slightly more over the short term than a pulse of emission warms it, but less over the long term." I can see why you get more cooling in short term (cause of the logarithm). What I can't rationlise is why you get less over the long-term (which feedback is causing the issue, it's the concentration feedback right because the climate feedback makes the sinks even more efficient)?

2. It seems computationally expensive to run all 10 000 combinations for all experiments, before then throwing a bunch away based on a few experiments (mainly ssp585 and 1pctCO2 I'm guessing). Would it not be faster to just run everything for ssp585 first, do some exclusions. Then 1pctCO2 (with whatever configurations remain), do more exclusions. Then run the rest of the experiments? That would save almost a factor of 10 in computing time or do I miss something?

3. page 6, line 169: It is sort of surprising that the fit with cumulative net ocean carbon flux gets worse from unconstrained to constrained. It would be great to have more text on why this could be. Could it be because there is a lack of covariance between likelihoods in the current calculation? It seems like the cumulative CO2 metrics get very high weight (because they're used 4 times effectively as there's 4 scenarios), although that should make the cumulative CO2 agreement better, not worse.

4. page 7, line 214: "OSCAR's overall ability to simulate the RF of short-lived species compares well with the IPCC AR5 values." The methane RF is outside the shown AR5 range, that doesn't seem like comparing well? It's also surprising that the methane RF is lower than AR5 (given Etminan revised it even further upwards). It's also not clear to me how total ERF agrees with AR5 (the lower bias from CH4 and aerosols seems to large to be neatly cancelled by the high bias of tropospheric ozone).

5. page 10, line 318: "the unconstrained carbon cycle of OSCAR v3.1 is well in line with CMIP exercices". Should 'unconstrained' → 'constrained'? Also, while Table 3 sort of supports this, Figure 1 shows pretty clear differences in cumulative compatible CO2 emissions between OSCAR and the constraint it is targeting so I'm not sure I would use 'well in line'.

6. Standard deviation seems inappropriate for skew distributions, can you show plumes of 5-95% and report 5th and 95th percentiles where distributions are skewed (e.g. aerosol ERF) and the appropriate data is available?

7. A curiosity, where is OSCAR's wildfire modelling best described?

**Figures**

Figure 3 caption: 'middle panels' → 'right panels' and 'right panels' → 'middle panels' ?

Figure 3: "Pearson's moment coefficient of skewness are provided in the legend" I can't see anything, am I missing something?

Figure 6: the arrows don't really help with knowing which direction things are moving. It's possible to intuit, but fixing the arrows would be preferable.

Figure 7: 'breached' → 'branched' ? 'Year after breach' → 'Year after zero emissions' ?

**Technical corrections**

page 1, line 15: comma missing, should read 'such a model, the newest'

page 1, line 15: 'observations from ESMs'? Can an ESM produce observations?

page 1, line 18: missing 'is' before 'that'

page 1, line 19: 'unstability' → 'instability' (and throughout)

page 2, line 41: 'relative' → 'relatively'

page 2, line 42: 'diagnose' → 'diagnosis'

page 2, line 48: 'increase' → 'increases'

page 2, line 54: 'some of' → 'some'

page 2, line 61: 'the DECK' → 'the CMIP6 DECK'

page 2, line 64: 'to the solar' → 'to solar'

page 2, line 67: 'exercice' → 'exercise'

page 3, line 74: 'and meant' → 'and is meant'

page 3, line 74: '(Gasser et al., 2017)' → 'Gasser et al. (2017)'

page 3, line 74: '(Gasser et al., 2020)' → 'Gasser et al. (2020)' (this referencing error is repeated in multiple places throughout)

page 3, line 74: 'We pinpoint that v3.1 is still calibrated on CMIP5 ESMs, then not meant to emulate CMIP6 models.' What does this mean?

page 3, line 77: 'It means that' → 'As a result,' (one suggestion, other phrasings would also work but the current formulation is clunky)

page 3, line 86: 'and with an' → 'with'

page 4, line 104: 'calibrated on' → 'calibrated to'

page 4, line 120: Should it read 'AR6 volcanic radiative forcing'? If no, have you used AR5 volcanic forcing with CMIP6 experiments, or what is going on?

page 5, line 127: delete 'may'? Or is there something else that can happen that hasn't been described?

page 5, line 128: As above, why do you use may? Are there other options? If yes, they should be described. If no, 'may' can be deleted.

page 5, line 130: 'represent' → 'represents'

page 5, line 149: "We acknowledge that when a significant fraction of the configurations is excluded, confidence in our model's result is lowered, but such a limitation of the validity domain is inherent to reduced-complexity models." Why does this lower confidence? It seems entirely normal for things to explode if you

put in particularly exotic parameter combinations (particularly in a non-linear model like OSCAR). It'd suggest deleting or rephrasing this line.

page 6, line 173: "the model returns after constraining 0.55 K", there is a word missing here somewhere, what does this mean?

page 7, line 192: The text says IPCC AR5 has a standard deviation of 18 PgC, but in the figure it looks more like 35 PgC. Can you please check as this impacts the impression of how close (or not) things are?

page 7, line 193: 'constrain' → 'constraint'

page 7, line 208: Which value do you end up plotting? AR5 rel. to 1850, rel. to 1750, rel. to 1850 using CMIP6 concentrations?

page 8, line 224: Could you clarify (perhaps earlier in the text) which experiment was used for the constraining? I'm guessing concentration-driven historical?

page 8, line 228: How do you come to the conclusion about ocean heat content? You only show one observational data point and it doesn't overlap with the timeseries from OSCAR.

page 8, line 226: "we note differences caused by volcanic eruptions" There don't seem to be any plots or other evidence to help evaluate the size of these differences. Can you please provide some move information on what you're referring to here please?

page 8, line 237: Delete "These results are shown in Figure 3." Sentences such as this aren't needed, just refer to the figure where relevent.

page 8, line 240: 'In regard of' → 'In regard to'

page 8, line 246: 'participates in' → 'contributes to'

page 8, line 252: 'follows the' → 'follows'

page 9, line 258: 'at the end of the 1,000 years' → 'at year 1,000'

page 9, line 259: 'about' → 'related to'

page 9, line 265: "The higher values for the ECS from some CMIP6 models are significantly reduced when constraining (Nijsse et al., 2020; Bonnet et al., 2021), with ECS even lower – 1.38K with a likely range of 1.3-2.1K – than those shown by OSCAR here." There are some words missing here (and maybe one of the ECS is meant to be TCR) so I couldn't follow this sentence.

page 9, line 268: "provided by OSCAR remain consistent with the litterature" Can you please include the OSCAR values and literature values so it's easy to judge whether these are indeed consistent? Or more clearly refer to Table 2 in the text so the reader knows where to look.

page 9, line 269: 'unconstrained' → 'constrained'? Again, please provide values so the reader can judge the level of consistency.

page 9, line 286: 'are strongly' → 'are more strongly'

page 10, line 293: 'on these' → 'in these'

page 10, line 320: "In any case, this suggests that our carbon cycle may be too optimistic", meaning that the uptake is too high for the same level of warming?

page 11, line 340: 'of the hydrological' → 'on the hydrological'

page 11, line 343: 'fully coupled ESM' → 'emissions-driven reduced complexity model', OSCAR isn't an ESM

page 11, line 346: 'to its' → 'at its'

page 11, line 350: 'at the same rath than the ramp-up period' → 'that is the reverse of the ramp-up period'

page 11, line 351: 'over' → 'for a further'

page 12, line 357: 'return within' → 'return below' ?

page 12, line 367: 'pinpoint' → 'note'

page 12, line 376: 'aims at' → 'aims to'

page 12, line 384: 'First, the ZEC in 385 branched experiments is systematically lower than the one in bell experiments.'. I see the opposite in Figure 8. The branched experiments are all higher in the corresponding year e.g. the branched 2000 PgC peaks around 0.2C, whereas the bell 2000 PgC never gets that high.

page 13, line 399: 'group of forcing' → 'group of forcings'

page 13, line 412: 'For the simulations under natural forcings, the range from the constrained OSCAR is smaller than the ones from (Gillett et al., 2021), which may suggest an over-constraining.' Is the range from Gillet quoted or am I missing something? It could also be that Gillet is under-constrained?

page 13, line 416: 'cannot conclude as to' → 'cannot comment on'

page 14, line 447: 'preindustrial' → 'quasi-preindustrial'

page 15, line 463: "Both shifting cultivation and wood harvest have no impact at all on the land sink, by construction of their formulation in OSCAR (Gasser et al., 2020)." Is this what we would expect based on first principles? Or is it a quirk of OSCAR?

page 15, line 466: should 'increases' be 'decreases'? Please clarify the experimental design and why it is one way or another (my intuition was that having grasslands instead of croplands would increase the sink as written but I can't see how to infer this from Figure 9, where negative numbers imply a reduction in the sink).

page 15, line 472: 'an land' → 'a land'

page 15, line 481: 'increase of -17 $\pm$ 13 PgC in the land' $\rightarrow$ 'decrease of 17 +/- 13 PgC in the land' would be clearer

page 17, line 524: 'warming' $\rightarrow$ 'causing less effective radiative forcing' because you're not talking about warming

page 18, line 552: 'ability of properly isolating' $\rightarrow$ 'ability to properly isolate'

page 19, line 582: "four tested scenarios, respectively", which order (SSP585-RCP585 first or SSP126-RCP26 first)?

page 19, line 587: 'Follows' $\rightarrow$ 'What follows is'

page 20, line 612: delete 'would'

page 38, line 1069: 'cases' $\rightarrow$ 'case'

---

## Author Comment (AC1)

**Response to Anonymous Referee 1 for the manuscript**

**CMIP6 simulations with the compact Earth system model OSCAR v3.1**

Yann Quilcaille[1*], Thomas Gasser[2], Philippe Ciais[3], Olivier Boucher[4]

[1] International Institute for Applied Systems Analysis (IIASA), 2361 Laxenburg, Austria;
[*] now at Institute for Atmospheric and Climate Science, Department of Environmental Systems Sciences, ETH Zürich, Zürich, Switzerland
[2] International Institute for Applied Systems Analysis (IIASA), 2361 Laxenburg, Austria
[3] Laboratoire des Sciences du Climat et de l'Environnement, LSCE/IPSL, Université Paris-Saclay, CEA – CNRS – UVSQ, 91191 Gif-sur-Yvette, France
[4] Institut Pierre-Simon Laplace, Sorbonne Université, CNRS 75252 Paris, France

Correspondence to: Yann Quilcaille (yann.quilcaille@env.ethz.ch)

We thank a lot the Anonymous Referee #1 for the comments. We think that their integration of these comments have been important to improve the quality of this manuscript. In the following response, the original answer is in *black italic* while the answer is in green.

Here is the summary of the modifications brought to the text:

- Improvement in the structure of the manuscript: reorganized the sections on the diagnostic of the model, moved to the appendix the sections on the behavior of the model
- More discussion in the experimental setup:
    o Description of OSCAR, with more emphasis brought on descriptions existing in the literature
    o New conceptual figure for description of the model
    o Post-processing of OSCAR more detailed
    o New conceptual figure for description of this framework, discussion of its limits
- More discussion in the sections on the diagnostic of the model
- Edit of the abstract
- More active formulations, proofread
- Correction of the figures to integrate the labelling of panels

***General comments***
*Quilcaille et al. present a summary of a number of CMIP6-style simulations performed with the reduced complexity earth system model OSCARv3.1. They briefly describe the model, how they constrained it to create probabilistic results and then go through the results of their numerous experiments.*
*The paper is clearly the result of a massive amount of effort, and having this description of OSCAR in the literature is beneficial. My issue with the current presentation is that it lacks any punch. There are so many results, and comparatively so little discussion, that it's hard to know what I'm meant to take away from this apart from, "We ran our model heaps". That's not to say that there aren't really interesting results in the paper, it's just that they're overly hard to find.*
We thank the Anonymous Referee 1 for this comment. We agree that the take-home messages of this manuscript are hard to identify.

As remarked, the amount of results for relatively little discussion makes it harder to read. To deal with the lack of discussion of the results, we now discuss into more details the sections concerning the diagnosis of the model.

Although the model is of "reduced complexity", there are many aspects of the model to cover, and we have been thorough. The take-home messages are indeed difficult to find, that is why we have also rearranged the structure of the manuscript, with results & discussions rearranged in two groups. The first group concerns the diagnosis of the model, and has been kept. The second one gathers results where elements of comparison are missing, but that still inform about the behavior of the model. These sections have been moved to the appendix.

We hope that these two categories of changes would provide a better guidance to the reader on what they are looking for about OSCAR.

*I think the paper would benefit greatly from improved focus. Many of the experiments require extensive discussion and further exploration. There doesn't appear to be space for all of them in this paper. For those that cannot be fully explored, I think it is better to save them for future papers where they can be explored appropriately rather than having partial explorations (also because I don't think partial explorations belong in the scientific literature).*
We agree with you that there is no space for a full discussion about all the aspects of this reduced complexity Earth system model. It would increase the size of the manuscript that is already more than 50 pages.

You suggest to save them for future papers, but is very unlikely that we will have the time to write papers on that many topics. Besides, it is even more unlikely that these would be scientifically interesting in their own right. Most of these sections that you identify are actually elements illustrating the behavior of OSCAR.

We think that this second group, now in the appendix, is still interesting. Showing how OSCAR behaves in different situations gives a sense of what to expect of the model to the reader. Besides, as far as we know, OSCAR is the only reduced-complexity Earth system model of RCMIP to provide details about the runs it performed for RCMIP. Nevertheless, we are willing to remove this appendix if the Editor considers that these results are unnecessary.

***Major concerns***
***Lack of focus***
*As discussed above, the paper lacks focus. Results are presented from experiment after experiment, with no space to actually explore their implications or what to make of them. Obvious examples are Sections 7.2, 7.3 and 7.5. However, many of the sections left me wondering, "What is the point?"*
As answered above, we have reorganized the manuscript to give a more appropriate structure and highlight the results. We have also given more space to discuss the implications of these results when appropriate.

*This is unfortunate, because many of the results are very interesting. For example, it is surprising that the ZEC is much higher for 2000 PgC experiments but overall warming doesn't have the same non-linearity. However, there is no further exploration of this.*
This part is now discussed into more details.

*Similarly, "the carbon stocks still increase in G6solar, even more than in ssp585 thanks to the lower GSAT and despite lower global precipitation." Is this what we would expect? Or does this point to a clear limitation of the model if we have less rain but somehow more carbon stocks?*
Thanks to the new structure and new discussion there, this part is better explained.

*Eight different spin ups are done: how do they compare? What does this tell us about the way we make climate projections and any potential bias in the CMIP-style of doing things.*
Although the question that you raise is interesting, we will not compare these spin-ups in this paper. As you noticed, we have already enough results and had not enough discussion, adding more results would not move this balance in the right direction. Furthermore, we consider that reduced-complexity Earth system models are not the proper tool to evaluate such biases, because such models are much less sensitive to the spin up itself than full-fledged Earth system models.

*The authors also write, "Our results cannot be compared to the final CDRMIP results yet, for they are unpublished, but they are consistent with those obtained with a model of intermediate complexity (Zickfeld et al., 2021)." However, they have a great tool to evaluate the questions: if they know how much to trust their model (which they should at the end of this evaluation), then they don't need to wait for the CDRMIP results and could write a great (separate) paper on their results now.*

As you suggest, it could do a great paper, and you are right in saying that it should be a separate one: this is not the objective of this manuscript. However, it is highly unlikely that we will have the time to write a paper dedicated to this question. However, we are of course open to collaborations for comparison and analysis, for instance in projects such as CDRMIP or ZECMIP.

*I would recommend the authors reconsider which results to present in this paper, which belong in their own paper and which are best left out. This would probably significantly reduce the length of the paper. It would also improve the abstract, which is currently too long and contains too much detail (it could be re-written to just focus on key points: Emulators are needed, emulators need to be validated, here we examine OSCAR, strengths are X, weaknesses are Y, last sentence could stay as is).*

As explained previously, this manuscript had two types of results: those about the diagnosis of the model, with elements of comparison to the literature, and those about the behavior of the model. We changed the structure of the paper accordingly, in a much clearer way, to guide the reader. Besides, we added discussion where it was needed. We think that these modifications would compensate for the aforementioned issues.

We have also edited the abstract, following your suggestions.

**Writing**
*The writing is very slow, i.e. it doesn't always make clear what the point is. I think this is partly due to a lack of focus as discussed above. It's partly due to being repetitive (the point about the need for validation is mentioned three times in the first paragraph). However, I think it is also partly due to phrasing. It might help to swap phrases like, "As illustrated in Table 4, OSCAR v3.1 estimates a ZEC (in the reference case of the esm-1pct-brch-1000PgC experiment) that is within the range of ZECMIP (Macdougall et al., 2020), although the long-term decrease seems to happen later in OSCAR." with more active formulations (that move all the table and figure references to parentheses) like "OSCAR v3.1 estimates a ZEC (in the reference case of the esm-1pct-brch-1000PgC experiment) that is within the range of ZECMIP (Macdougall et al., 2020), although the long-term decrease seems to happen later in OSCAR (Table 4)." (Yes, I acknowledge the irony of giving advice about punchy writing when my review is probably slightly rambling.) The paper is also in need of a proofread. Reading it this time was overly difficult due to typing and other phrasing errors. This will take some effort, but it will greatly improve the experience for the reader.*

We thank a lot the Referee for this comment. We have gone through the paper to work on these aspects.

**Vague claims of goodness**
*The paper has some very vague claims of goodness. Two examples, "It reproduces the responses of complex ESMs, for all aspects of the Earth system.", and, "the resulting quantitative behaviour of OSCAR remains largely satisfactory". Both these claims are vague and subjective. I would simply remove them and all others like them from the paper, the reader can judge the quality for themselves based on the results (and likely, the 'good enough' level will change depending on the application of interest).*

Depending on the context, we have either removed the claim if no points of comparison were possible, or we have given more details for a better comparison. This is in line with the correction for more discussions where needed.

---

## Author Comment (AC2)

**Response to Anonymous Referee 2 for the manuscript**

**CMIP6 simulations with the compact Earth system model OSCAR v3.1**

Yann Quilcaille[1*], Thomas Gasser[2], Philippe Ciais[3], Olivier Boucher[4]

[1] International Institute for Applied Systems Analysis (IIASA), 2361 Laxenburg, Austria;
* now at Institute for Atmospheric and Climate Science, Department of Environmental Systems Sciences, ETH Zürich, Zürich, Switzerland
[2] International Institute for Applied Systems Analysis (IIASA), 2361 Laxenburg, Austria
[3] Laboratoire des Sciences du Climat et de l'Environnement, LSCE/IPSL, Université Paris-Saclay, CEA – CNRS – UVSQ, 91191 Gif-sur-Yvette, France
[4] Institut Pierre-Simon Laplace, Sorbonne Université, CNRS 75252 Paris, France

Correspondence to: Yann Quilcaille (yann.quilcaille@env.ethz.ch)

We are grateful to the Anonymous Referee #2 for the comments. After integration of these comments, the quality of this manuscript has improved. In the following response, the original answer is in *black italic* while the answer is in green.

Here is the summary of the modifications brought to the text:

- Improvement in the structure of the manuscript: reorganized the sections on the diagnostic of the model, moved to the appendix the sections on the behavior of the model
- More discussion in the experimental setup:
    - Description of OSCAR, with more emphasis brought on descriptions existing in the literature
    - New conceptual figure for description of the model
    - Post-processing of OSCAR more detailed
    - New conceptual figure for description of this framework, discussion of its limits
- More discussion in the sections on the diagnostic of the model
- Edit of the abstract
- More active formulations, proofread
- Correction of the figures to integrate the labelling of panels

*Quilcaille et al describe a large number of simulations conducted with the OSCAR simple climate model, covering some fraction of the CMIP6 experimental design. The paper briefly describes the model and the calibration strategy, and then discusses a number of applied cases.*

*Clearly a large amount of work has gone into this paper, singlehandledly simulating a large fraction of the CMIP6 experimental design in a single study. However, a side-effect of this is that the paper is somewhat less than rigorous in describing the calibration process and in providing a detailed assessment of the applied experiments in the context of available ESM simulations.*
We thank the Anonymous Referee for this comment. We have provided more details on the probabilistic setup. We highlight that this paper is not about the calibration of OSCAR, because this model had already been calibrated before, as described in Gasser et al, 2017 (https://doi.org/10.5194/gmd-10-271-2017).
We have as well given more space to the discussion of the results where needed and changed the structure, to bring more clarity on the take-home messages.

*It is also sloppily written in places - with a number of spelling errors and inprecise language. Figures require axis labels and subplot labels throughout.*
A thorough proofreading of the manuscript has improved the spelling and language.

Regarding the figures, it is a choice justified by the clarity of the figures. For instance, subplots sharing the same time axis do not need to have "year" and the corresponding tick labels repeated over and over again. A good example is figure 9 for the LUMIP experiments: all columns share the same time axis, then only the bottom row has the axis label written. We think that changing all these figures would add a lot of text and details for information already present, thus decreasing the readability of the figures.
Regarding the labeling of the subplots, we have edited the figures accordingly to the GMD guidelines.

*My recommendation is that the scope of the paper is reduced - but greater emphasis is placed on a detailed description of the calibration process, and the senstivities of simple scenarion projections to aspects of that calibration process. Additional experiments (SRM, CDR etc) - are secondary and could be covered in follow-up dedicated studies when the fundamental probabilistic setup is well defined.*
We have changed the whole structure for the results, identifying two groups of sections, one about the diagnosis of the model and another about its behavior (SRM, CDR, etc). As you suggest, this second part is indeed secondary. It has been kept as an appendix, because we consider that it is still important to the reader, as a showcase of the behavior of the model. This part illustrates what to expect of the model, which is what some readers are looking for when reading about a model. We highlight that no other reduced complexity Earth system model has done such a thorough analysis before, and such a paper could be a first step towards better descriptions.

Of course, this appendix about the behavior could be removed if the Editor decides that it is not relevant. However, it is highly unlikely that the deleted material would become future papers, because we will not have the time for that many additional studies.

*Major issues:*
*1 - More detail is needed on the basic model structure. This version of the model is not documented in the literature, and the first section needs to give a basic overview of the level of complexity being represented. A conceptual figure illustrating the number of domains, and how they interact, would be appreciated.*
We thank the Referee for this comment that has improved the clarity of the manuscript. We have to disagree on the documentation of this version, however. The second sentence of the first paragraph on the description of this version explicitly explains where the version 2.2 is described and where to read about the difference from version 2.2 to version 3.1.

Even though OSCAR is a "reduced complexity" Earth system model, giving a basic overview of what is represented is no easy task. The 4 following paragraphs were actually aiming at that, but your comment shows that it was not enough. For this reason, we introduce a new figure, providing a conceptual overview of the model. Although this is only a conceptual figure, without the details of all equations, datasets and parameters, it provides a good perspective on the level of complexity of the components and their interactions. Of course, we remind the sources for the detailed descriptions of the versions of OSCAR in the caption of the figure.

*2 - Perhaps the key aspect of this paper is the effect of observational constraints on projected future climate. However, this is covered quite briefly, and the many degrees of freedom in the calibration process are not comprehensively explored. How do different observations constrain projected warming independently and combined? How are constraints objectively combined? How are prior distributions decided? The paper refers back to Gasser (2017), but the addition of additional constraints in the present paper requires a more detailed description.*
We thank the Referee for this comment. The use of observational constraints is important in this paper, but we do not consider this aspect as the main scope of the paper. We highlight that this is constraining, and not calibrating. The calibration concerns the parameters for the equations, which was described in Gasser et al (2017) and Gasser et al (2020). Regarding the constraining, this followed the coordinated effort of RCMIP (phase 2), but going into the details of the (infinite) possibilities of doing such constraining is an entirely different paper in itself. Here, we acknowledge the limits of our approach and the need to investigate this further for future use with OSCAR.

However, what was done for RCMIP is done and will not change, therefore we report in this paper the performance of this precise version of the model.

The manuscript refers to Gasser et al (2017) for the model description, the parameters and the probabilistic framework. However, it is Gasser et al (2020) that provided information on the update of the model and the use of observational constraints.

Nevertheless, we have added details on the limit on how the observational constraints have been chosen and used. We hope that the addition of the new figure and of more details on the post-processing will bring the sought clarifications.

*3 - The language is often far too vague on key details, like exactly how CMIP5 data were used: (e.g. . "The preindustrial state of the land carbon cycle is calibrated against TRENDYv7 and its transient response to CO2 and climate is calibrated against CMIP5 models"). The key thing to communicate in this paper is exactly what data and models were used to calibrate OSCAR, what parameters are being calibrated and what is the sensitivity of the model to each piece of information.*
We thank the Referee for this interesting comment. As mentioned earlier, all of these aspects have already been published in the reference paper of OSCAR that was cited throughout all the manuscript. In Gasser et al (2017), 49 pages bring the information that you are asking for. As explained earlier, OSCAR v2.2 is described in Gasser et al (2017) while the changes from v2.2 to v3.1 are explained in Gasser et al (2020). In this evaluation paper, no additional or different calibration of the parameters are performed: the model is used in its fixed version 3.1.

However, following your comment, we decided to give more visibility to these explanations in the first paragraph. Besides, we have gone through the text to improve the language.

*3 - The exclusion of unstable parameter combinations is understandable - but the conditions for instability need to be more objectively quantified. Plots illustrating the instabilities, and conditions for exclusion would be appreciated, together with a process assessment of why they occur and whether the exclusion process might be biasing the observationally constrained distributions.*
We thank the Referee for this interesting comment. We acknowledge that the exclusion process may bias our conclusions. We are now giving more visibility to this aspect in the manuscript.

We had explained that these exclusions were due to the instability of the ocean carbon cycle, hence the described method. Further analysis of this aspect could be performed along the improvement of this module that we mentioned. According to your comment, we are giving more details about the exclusion, in the limit of what is feasible. We explain in more details how we proceed, what it means for the model.

*4 - The joint constraint on cumulative carbon uptake and warming results in a tightening of a prior distribution, which was already narrower than the observational uncertainty (Figure 1 (b?) - sublabels are required!). This tightening occurs due to the effect of the constraint on warming, but means that the model ensemble is not plausible sampling solutions with low carbon uptake. This is a potential bias in the assessment of the model distributions, unless we have perfect confidence in the model structure - which we don't. The constraining approach would benefit from having a parameter which allowed for model imperfection in calibration (see McNeall 2016 for an example of this calibration problem and Williamson 2019 as an example of a statistical framework to address it).*
We agree with the Referee that the constraining approach could be improved, and the papers that you cite would be of a great help in this sense. We thank the reviewer for providing an interesting lead for further development of the model. We now acknowledge this limit in the paper and cite these papers as a way to overcome it.

*5 - It would be useful, thoughout, for plots to show CMIP5 and CMIP6 distributions where available in addition to the OSCAR distributions.*

We thank the Referee for this comment. We are adding comparison to the existing literature wherever possible, be it from CMIP5 or CMIP6, or directly from the 6th Assessment Report of the IPCC. It concerns the sections treating the diagnosis of OSCAR. Yet, the section treating the behavior of OSCAR is more for illustration purposes.

We have however elected not to overburden the figures with such comparison, and to limit this type of information to the text or to tables.

---

## Author Comment (AC3)

**Response to the Editor for the manuscript**

**CMIP6 simulations with the compact Earth system model OSCAR v3.1**

Yann Quilcaille[1*], Thomas Gasser[2], Philippe Ciais[3], Olivier Boucher[4]

[1] International Institute for Applied Systems Analysis (IIASA), 2361 Laxenburg, Austria;
* now at Institute for Atmospheric and Climate Science, Department of Environmental Systems Sciences, ETH Zürich, Zürich, Switzerland
[2] International Institute for Applied Systems Analysis (IIASA), 2361 Laxenburg, Austria
[3] Laboratoire des Sciences du Climat et de l'Environnement, LSCE/IPSL, Université Paris-Saclay, CEA – CNRS – UVSQ, 91191 Gif-sur-Yvette, France
[4] Institut Pierre-Simon Laplace, Sorbonne Université, CNRS 75252 Paris, France

Correspondence to: Yann Quilcaille (yann.quilcaille@env.ethz.ch)

We are very grateful to the Editor for allowing these discussions. In the following response, the original answer is in *black italic* while the answer is in green. Here is the summary of the modifications brought to the text:

- Improvement in the structure of the manuscript: reorganized the sections on the diagnostic of the model, moved to the appendix the sections on the behavior of the model
- More discussion in the experimental setup:
  - Description of OSCAR, with more emphasis brought on descriptions existing in the literature
  - New conceptual figure for description of the model
  - Post-processing of OSCAR more detailed
  - New conceptual figure for description of this framework, discussion of its limits
- More discussion in the sections on the diagnostic of the model
- Edit of the abstract
- More active formulations, proofread
- Correction of the figures to integrate the labelling of panels

*The manuscript has now been reviewed by two referees. I agree with their view that the paper is of general interest, but presents to large a set of simulations without adequate interpretation and discussion of results.*

We acknowledge that this manuscript presents many results, covering numerous aspects of this model and experiments. Because of the amount of results, we did not discuss in details all results.

We consider that some sections require discussion on the results, and they are those that concern the diagnostic of the model. The structure of the manuscript did not allow to identify these sections easily. This is why we have rearranged the main structure of the manuscript into the experimental setup, the diagnosis of the model and the behavior of the model. The sections concerning the diagnosis of the model now present more discussion of the results.

To comply with the recommendations of the reviewers, we moved the sections on the behavior of the model to the appendix. We deem them as useful for many readers, because they describe the response of the model in situations that the sole diagnostic of the model does not allow to imagine. So far, Reduced-Complexity Earth System Models have shown their outputs only for usual experiments,

but did not show their behaviors in all these situations. That is why we consider that it would still be a valuable input to the literature.

Of course, if you consider that this section on the behavior of the model would not be of interest to the readers, we would agree on deleting it. However, we highlight that we would not be able to invest the time to transform these deleted results into future studies, hence they would be lost.

*Is OSCAR 3.1 already fully documented from Gasser et al. (2017) and Gasser et al. (2020), including calibration? If not, more emphasis needs be spent on the model description part.*

The description of the model OSCAR v3.1 and its calibration are fully documented from these 2 papers. Although it was written in the section on the description of the model, it was not clear enough. That is why we have brought more details on several aspects of the experimental setup:
- New conceptual figure, describing OSCAR v3.1. We tried to keep it as conceptual as possible, but still ended up with +60 boxes.
- More emphasis brought on the description
- New conceptual figure for the post-processing of OSCAR
- Post-processing of OSCAR more detailed, acknowledging its limits as suggested by Reviewer 2.

Please prepare a revised manuscript of the paper addressing this main concern and the other reviewer comments, as well as a point-by-point response.

All comments have been answered in the other files. We will send you soon the revised manuscript.

---

## Author Response (AR1)

**Response to the Editor for the manuscript**

**CMIP6 simulations with the compact Earth system model OSCAR v3.1**

Yann Quilcaille[1][*], Thomas Gasser[2], Philippe Ciais[3], Olivier Boucher[4]

[1] International Institute for Applied Systems Analysis (IIASA), 2361 Laxenburg, Austria;
[*] now at Institute for Atmospheric and Climate Science, Department of Environmental Systems Sciences, ETH Zürich, Zürich, Switzerland
[2] International Institute for Applied Systems Analysis (IIASA), 2361 Laxenburg, Austria
[3] Laboratoire des Sciences du Climat et de l'Environnement, LSCE/IPSL, Université Paris-Saclay, CEA – CNRS – UVSQ, 91191 Gif-sur-Yvette, France
[4] Institut Pierre-Simon Laplace, Sorbonne Université, CNRS 75252 Paris, France

Correspondence to: Yann Quilcaille (yann.quilcaille@env.ethz.ch)

Thank you very much for accepting our propositions to adapt the manuscript. The corresponding changes have been made, thus improving the clarity of the manuscript. Please note that the file for tracked changes does not integrate the change in structure, to improve the readability of the corresponding file. All other changes are tracked.

---

## Referee Report (RR1)

**General comments**

Quilcaille et al. have revised their paper to enhance the focus on model evaluation, leaving pure presentation of model behaviour for the appendix. I think these changes have improved the manuscript. The addition of Figure 1 was also very helpful for me.

My major concerns focus on a few key areas, many of which echo earlier comments on the manuscript. I think these can be addressed. I also still think the point of the paper could be made clearer (there were fewer tracked changes than I was expecting to see). Is it not: we have used OSCAR v3.1 in a few places already, here we provide a thorough evaluation of its behaviour over a number of experiments where we have something to compare against, here are the levels of agreement (quantified)? If the paper just stuck to telling this narrative, I think it would be much easier to read.

I would also note that many of the other reviewer's comments put a pretty high expectation on the authors. In my opinion, many of the questions asked about particular details and choices related to calibration are better explained by the code accompanying the paper (rather than duplicating this information in the paper) or in standalone papers. Adding such things into a pure evaluation paper (whatever that is worth, see comments below) makes it very hard to have focus.

Overall, I think the paper now achieves its aim of evaluating the behaviour of OSCARv3.1. However, I do think it could be greatly improved in terms of presentation and clarity.

**Major concerns**

**Vague claims of goodness**

The vague claims of goodness persist in this version of the manuscript (even in the abstract). Where they appear, they read like the authors want to be able to say, "OSCAR is good", which is particularly odd, because the authors are very honest about the limitations of their model in many other parts. Again, I would just remove any sentence that uses a subjective judgement, such as 'good' or 'satisfactory'. Just tell the reader what the difference is and they can decide what is good enough based on their own situation.

**Behavioural description**

The authors have retained their section that focuses purely on behaviour of the model, albeit as an appendix. I can see why they want to keep this section, but I have some further thoughts about this.

The first is this. In the revision process, the authors make statements like the following, "However, we highlight that we would not be able to invest the time to transform these deleted results into future studies, hence they would be lost." The implication is that this is the only chance to publish them. My

issue with this statement is that, by saying, "We won't have time", the authors are implicitly saying, "We won't make time". Put another way, the authors are saying that, "These results aren't interesting enough to be worth our time writing up". The problem with this is that it then raises the question, are these results worth anyone's time reading? I think it is ultimately an editorial question whether these pure documentation plots can be included in an appendix or not (they take up space and are disconnected from the main narrative of the manuscript, but you don't have to read them to understand the manuscript so they aren't a negative). However, I still struggle to see why plots of stuff, without any explanation of their implications, belong in the scientific literature (surely they are better captured as part of a tutorial on the model or the model's development repository, where they can be presented without any accompanying narrative?).

My second thought also follows from a comment by the authors, "We highlight that no other reduced complexity Earth system model has done such a thorough analysis before, and such a paper could be a first step towards better descriptions." I would agree with this (more or less) and I think it raises a fascinating question about how to document different model versions. The current practice of writing standalone manuscripts is clunky for a number of reasons. Firstly, a complete description of the model is not appropriate for any single manuscript so it never appears anywhere (rather, any user has to piece together the full picture from multiple papers). Secondly, description papers tend to be very long because they have to cover so much territory. Thirdly, they are very hard to write because they don't have an obvious narrative apart from, "Here is how the model looks/works" (and that narrative isn't very interesting to most people given models are for insight, not for numbers). Given that current practice is clunky, I would encourage the editors of GMD to give this question futher thought: How can model description papers be improved so that they are more useful for authors and readers alike? Are scientific papers even the right forum for such documentation given their focus on narrative and implications? Obviously these questions don't affect the publication of this paper, but given the authors' made the comments I thought I would reply.

**Minor concerns**

**Diagnosis vs. evaluation**

The authors refer to the new first section as diagnosis. This language seemed odd to me, I would have used the phrase evaluation because the authors seem to be evaluating the extent to which their model behaves in line with other available literature estimates over a range of experiments. An introductory paragraph at the start of section 3 (before the section 3.1 header) that re-clarifies the point of this section would be helpful (given how long section 2 is).

**Reproducibility**

The paper's reproducibility would be greatly enhanced if it was clear where an interested party could access the code that sits behind it, particularly the code related to constraining OSCAR. Having the model code available open-source is good, but it isn't enough to actually reproduce the paper's results by itself and the descriptions given in the paper are certainly not enough to reproduce the study by themselves.

**Technical corrections**

A selection are listed below, but I would note that the paper is still in need of a good proofread as many of the sentences are still missing words and use odd phrases, which makes reading the paper much harder than it needs to be.

page 1, line 12: 'spatial' → 'spatial and temporal' (noting that ESMs often run on sub-daily timesteps)

page 1, line 16-18: 'Overall, OSCAR v3.1 shows good agreement with observations, ESMs and emerging properties. It reproduces the responses of complex ESMs, for all aspects of the Earth system.' Sentence is meaningless without quantification, either delete or add numbers (and remove subjective measures of goodness like 'good')

page 2, line 43: 'is increased' → 'is also increased'

page 2, line 48: 'of CMIP6' → 'CMIP6'

page 2, line 56: 'to evaluate' → 'are used to evaluate'

page 3, line 65: 'meant' → 'is meant'

page 5, line 143: 'As illustrated in Figure 2', I don't see this at all in figure 2...

page 5, line 169: delete 'are used'

page 6, line 190: Suggest adding words like 'following' before the reference to Mcneall. The references don't illustrate the point, but they point in a direction for further improvement.

page 14, line 430: 'on carbon' → 'under experiments that examine'

page 19, line 690: 'resulting quantitative behaviorbehaviour of OSCAR remains largely satisfactory', suggest removing all these vague assertions

---

## Author Response (AR2)

**Response to the Editor for the manuscript**

**CMIP6 simulations with the compact Earth system model OSCAR v3.1**

Yann Quilcaille[1*], Thomas Gasser[2], Philippe Ciais[3], Olivier Boucher[4]

[1] International Institute for Applied Systems Analysis (IIASA), 2361 Laxenburg, Austria;
* now at Institute for Atmospheric and Climate Science, Department of Environmental Systems Sciences, ETH Zürich, Zürich, Switzerland
[2] International Institute for Applied Systems Analysis (IIASA), 2361 Laxenburg, Austria
[3] Laboratoire des Sciences du Climat et de l'Environnement, LSCE/IPSL, Université Paris-Saclay, CEA – CNRS – UVSQ, 91191 Gif-sur-Yvette, France
[4] Institut Pierre-Simon Laplace, Sorbonne Université, CNRS 75252 Paris, France

Correspondence to: Yann Quilcaille (yann.quilcaille@env.ethz.ch)

We want to sincerely thank again the Editor for their approval of this manuscript. Further modifications were brought to the manuscript, following the comments of the referees. In the following response, the original answer is in *black italic* while the answer is in green.

Here is the summary of the modifications brought to the text:

- Improved narrative, focus on the results.
- Removed vague claims of goodness and more quantification.
- New appendix for post-processing.
- New repository for the code of OSCAR for replication.

*Thanks you for your revised version of the manuscript. Both reviewers appreciate the improvements on the earlier version. Still, reviewer #2 points to further room for improvement. In particular, I support her/his request to refine the framing and narrative of the paper, and would also ask you to address the other points.*
*Regarding the in-depth presentation of model behavior, I am fine with keeping it in the appendix.*
*I am looking to a revised version of your paper.*

The aforementioned modifications were brought to the manuscript, following Reviewer #2's recommendations. Specifically, the narrative & framing were improved with the following points:
- Although the section on post-treatment of the runs brings useful information for the understanding of the method, it was too long, with the risk to lose the reader. We moved a large part of this section to a new appendix.
- We went through the text to improve it, following the recommendation of Reviewer #2. Several modifications in the text were made to strengthen the narrative. In particular, the introduction has been modified along this line and a new paragraph was added on the recommendation of Reviewer #2 between section 3 and 3.1, serving as reminder.
- Reviewer #2 asked for more quantification. We noticed that the section 3.4 and the abstract were not adequately quantified, contrary to the other sections that benefit from the 5 tables. New additions were made there.

**Response to Anonymous Referee 1 for the manuscript**

**CMIP6 simulations with the compact Earth system model OSCAR v3.1**

Yann Quilcaille[1*], Thomas Gasser[2], Philippe Ciais[3], Olivier Boucher[4]

[1] International Institute for Applied Systems Analysis (IIASA), 2361 Laxenburg, Austria;
* now at Institute for Atmospheric and Climate Science, Department of Environmental Systems Sciences, ETH Zürich, Zürich, Switzerland
[2] International Institute for Applied Systems Analysis (IIASA), 2361 Laxenburg, Austria
[3] Laboratoire des Sciences du Climat et de l'Environnement, LSCE/IPSL, Université Paris-Saclay, CEA – CNRS – UVSQ, 91191 Gif-sur-Yvette, France
[4] Institut Pierre-Simon Laplace, Sorbonne Université, CNRS 75252 Paris, France

Correspondence to: Yann Quilcaille (yann.quilcaille@env.ethz.ch)

We want to sincerely thank the Anonymous Referee #1 for its approval on this manuscript. Further modifications were brought to the manuscript, following Anonymous Referee #2's comments. In the following response, the original answer is in *black italic* while the answer is in green.

Here is the summary of the modifications brought to the text:

- Improved narrative, focus on the results.
- Removed vague claims of goodness and more quantification.
- New appendix for post-processing.
- New repository for the code of OSCAR for replication.

*Thanks for the careful response to my original review. The improved focus and expanded details on the conditions required for exclusion of ensemble members is appreciated. I'm happy for the paper to be published in its current form.*
Thank you very much for your approval. As said before, Anonymous Referee #2 had several recommendations that we followed. In particular, the details on exclusion of ensemble members were moved to a new appendix to strengthen the narrative. We agree that the details that you suggested in the first round are useful for many readers, albeit it may lose some others. This is why we decided to keep them as the first appendix to focus on the main message in the main text.

**CMIP6 simulations with the compact Earth system model OSCAR v3.1**

Yann Quilcaille[1*], Thomas Gasser[2], Philippe Ciais[3], Olivier Boucher[4]

[1] International Institute for Applied Systems Analysis (IIASA), 2361 Laxenburg, Austria;
* now at Institute for Atmospheric and Climate Science, Department of Environmental Systems Sciences, ETH Zürich, Zürich, Switzerland
[2] International Institute for Applied Systems Analysis (IIASA), 2361 Laxenburg, Austria
[3] Laboratoire des Sciences du Climat et de l'Environnement, LSCE/IPSL, Université Paris-Saclay, CEA – CNRS – UVSQ, 91191 Gif-sur-Yvette, France
[4] Institut Pierre-Simon Laplace, Sorbonne Université, CNRS 75252 Paris, France

Correspondence to: Yann Quilcaille (yann.quilcaille@env.ethz.ch)

We want to sincerely thank the Anonymous Referee #2 for the comments. We have carefully read its report and integrated its comments. We deem that it has significantly improved the manuscript, one more time. In the following response, the original answer is in *black italic* while the answer is in green.

Here is the summary of the modifications brought to the text:

- Improved narrative, focus on the results.
- Removed vague claims of goodness and more quantification.
- New appendix for post-processing.
- New repository for the code of OSCAR for replication.

*Quilcaille et al. have revised their paper to enhance the focus on model evaluation, leaving pure presentation of model behaviour for the appendix. I think these changes have improved the manuscript. The addition of Figure 1 was also very helpful for me.*
Glad to read that it helped.

*My major concerns focus on a few key areas, many of which echo earlier comments on the manuscript. I think these can be addressed. I also still think the point of the paper could be made clearer (there were fewer tracked changes than I was expecting to see). Is it not: we have used OSCAR v3.1 in a few places already, here we provide a thorough evaluation of its behaviour over a number of experiments where we have something to compare against, here are the levels of agreement (quantified)? If the paper just stuck to telling this narrative, I think it would be much easier to read.*
Thank you for expressing this concern. In the first round of review, the structure has benefited from your comments. However, it is true that the narrative could still be improved. Three main axis have been used to improve the narrative:

- Although the section on post-treatment of the runs brings useful information for the understanding of the method, it was too long, with the risk to lose the reader. We moved a large part of this section to a new appendix.
- We went through the text to improve it, following your recommendation. Several modifications in the text were made to strengthen the narrative. In particular, the introduction has been modified along this line, and a new paragraph was added on your recommendation between section 3 and 3.1, serving as reminder.
- Regarding quantification, we noticed that the section 3.4 and the abstract were not adequately quantified, contrary to the other sections that benefit from the 5 tables. New additions were made there.

On a quick note, there were less tracked changes visible because the 6 pages of text and 8 pages of figure moved from the main text to the supplementary material were not integrated in the tracked changes, to make it easier to read. In this round of revision, all changes were tracked.

I would also note that many of the other reviewer's comments put a pretty high expectation on the authors. In my opinion, many of the questions asked about particular details and choices related to calibration are better explained by the code accompanying the paper (rather than duplicating this information in the paper) or in standalone papers. Adding such things into a pure evaluation paper (whatever that is worth, see comments below) makes it very hard to have focus.

Thank you. We agree that such technical questions would find a more exhaustive description in the code. However, the reasons for the choices made would not necessarily appear in the code. Besides, someone interested in these aspects may get a better understanding of the code by first reading the text and accompanying figures.

To balance the reviews, we moved the Figure 2 and a large part of the section 2.3 to the appendix. We hope that it would bring a better focus to the paper while allowing for readers interested in these details to find the sought information.

Overall, I think the paper now achieves its aim of evaluating the behaviour of OSCARv3.1. However, I do think it could be greatly improved in terms of presentation and clarity.

Thank you. We hope that this new version would have the adequate level of clarity.

The vague claims of goodness persist in this version of the manuscript (even in the abstract). Where they appear, they read like the authors want to be able to say, "OSCAR is good", which is particularly odd, because the authors are very honest about the limitations of their model in many other parts. Again, I would just remove any sentence that uses a subjective judgement, such as 'good' or 'satisfactory'. Just tell the reader what the difference is and they can decide what is good enough based on their own situation.

We acknowledge that the former round of corrections left some of these subjective judgements out. This is now corrected.

The authors have retained their section that focuses purely on behaviour of the model, albeit as an appendix. I can see why they want to keep this section, but I have some further thoughts about this. The first is this. In the revision process, the authors make statements like the following, "However, we highlight that we would not be able to invest the time to transform these deleted results into future studies, hence they would be lost." The implication is that this is the only chance to publish them. My issue with this statement is that, by saying, "We won't have time", the authors are implicitly saying, "We won't make time". Put another way, the authors are saying that, "These results aren't interesting enough to be worth our time writing up". The problem with this is that it then raises the question, are these results worth anyone's time reading? I think it is ultimately an editorial question whether these pure documentation plots can be included in an appendix or not (they take up space and are disconnected from the main narrative of the manuscript, but you don't have to read them to understand the manuscript so they aren't a negative). However, I still struggle to see why plots of stuff, without any explanation of their implications, belong in the scientific literature (surely they are better captured as part of a tutorial on the model or the model's development repository, where they can be presented without any accompanying narrative?).

When we had written that we would not have the time, it doesn't mean that we consider these results as "not interesting enough to be worth our time writing up". We do see scientific value in these results and foresee appealing papers that would make use of them, particularly on the reversibility of the Earth system using section A.1. The sole reason why we will not be able to make time for such papers is that in modern research, the priorities are set by funding and not by scientific interests. We have to focus on the projects that we are working for, even it means not publishing papers with scientific value. For instance, the main author is now in a different institute, paid on a different project, and the hours spent on this paper cannot be reported on this new project. Then no, it is not because the other papers would not have been interesting, but because we sincerely cannot take this time.

Furthermore, we would not like to have these results completely sacrificed. They would have a good fit in model intercomparison studies, as initially envisioned for the CDRMIP runs. Publishing these results is not enough as a stand-alone paper, but they may be of interest as contributions to other studies. This is why we still want to showcase OSCAR in the supplementary material of this manuscript.

Regarding the "pure documentation plots", I realized that I should have moved the figures of the appendix with their text. I apologize and corrected this mistake. Now, these plots are directly with their corresponding explanations in the appendix. As discussed in the first round of review, these sections on the behavior of the model may bring some insights to the readers interested in qualitative aspects of the model on these experiments, otherwise not shown.

My second thought also follows from a comment by the authors, "We highlight that no other reduced complexity Earth system model has done such a thorough analysis before, and such a paper could be a first step towards better descriptions." I would agree with this (more or less) and I think it raises a fascinating question about how to document different model versions. The current practice of writing standalone manuscripts is clunky for a number of reasons. Firstly, a complete description of the model is not appropriate for any single manuscript so it never appears anywhere (rather, any user has to piece together the full picture from multiple papers). Secondly, description papers tend to be very long because they have to cover so much territory. Thirdly, they are very hard to write because they don't have an obvious narrative apart from, "Here is how the model looks/works" (and that narrative isn't very interesting to most people given models are for insight, not for numbers). Given that current practice is clunky, I would encourage the editors of GMD to give this question futher thought: How can model description papers be improved so that they are more useful for authors and readers alike? Are scientific papers even the right forum for such documentation given their focus on narrative and implications? Obviously these questions don't affect the publication of this paper, but given the authors' made the comments I thought I would reply.

You are indeed raising a crucial point. This publication is trying to find its place in a "clunky" context, being the description of reduced complexity Earth system models and to a broader sense the description of models. Luckily, reduced complexity Earth system models are simpler than Earth system models, facilitating their full description in a single paper. This is the case for OSCAR, MAGICC, HECTOR and many others. However, it is true that later versions build upon the initial publication, meaning that reading on a model means piecing papers together. Automated wikis are of course possible, although it takes a significant input from a software perspective, and not all modelling teams are capable of that. Versioning tools like GitHub provide a good alternative, albeit less user-friendly as an automated wiki. Therefore OSCAR is on GitHub, although a systematic evaluation of the model's versions remain to be implemented. This paper is meant to provide a first milestone towards the comparison of future versions of OSCAR. For instance, adding new processes, improving the modeling of the ocean carbon sink, or changing the representation of aerosol chemistry in OSCAR would change its outputs. We consider that this paper would pave the way to a better understanding of what needs to be changed in the model and the effects of the ensuing changes. Of course, there are other solutions to improve model descriptions, each with their pros and cons. We hope that the path that we choose here would give readers a good overview of this model.

The authors refer to the new first section as diagnosis. This language seemed odd to me, I would have used the phrase evaluation because the authors seem to be evaluating the extent to which their model behaves in line with other available literature estimates over a range of experiments. An introductory paragraph at the start of section 3 (before the section 3.1 header) that re-clarifies the point of this section would be helpful (given how long section 2 is).

Thank you, we have edited the manuscript accordingly to your advice.

The paper's reproducibility would be greatly enhanced if it was clear where an interested party could access the code that sits behind it, particularly the code related to constraining OSCAR. Having the model code available open-source is good, but it isn't enough to actually reproduce the paper's results by itself and the descriptions given in the paper are certainly not enough to reproduce the study by themselves.

We have now created a new repository for the code used to run OSCAR, to find the diverging runs, to constrain OSCAR and for the plots produced here. Though, it comes with a warning that this code is very raw, hindering its readability. Again, it is the longer-term plan to integrate evaluation runs with the model's code on its GitHug repository, based on the work done for this paper, but this will take significant efforts.

Technical corrections:
A selection are listed below, but I would note that the paper is still in need of a good proofread as many of the sentences are still missing words and use odd phrases, which makes reading the paper much harder than it needs to be.

Thank you for such a careful reading of this manuscript, we have integrated these corrections and additional ones after proofreading. Would the reviewer point to any remaining issues, we would be very grateful.

---

## Author Response (AR3)

**Response to the Editor for the manuscript**

**CMIP6 simulations with the compact Earth system model OSCAR v3.1**

Yann Quilcaille[1*], Thomas Gasser[2], Philippe Ciais[3], Olivier Boucher[4]

[1] International Institute for Applied Systems Analysis (IIASA), 2361 Laxenburg, Austria;
[*] now at Institute for Atmospheric and Climate Science, Department of Environmental Systems Sciences, ETH Zürich, Zürich, Switzerland
[2] International Institute for Applied Systems Analysis (IIASA), 2361 Laxenburg, Austria
[3] Laboratoire des Sciences du Climat et de l'Environnement, LSCE/IPSL, Université Paris-Saclay, CEA – CNRS – UVSQ, 91191 Gif-sur-Yvette, France
[4] Institut Pierre-Simon Laplace, Sorbonne Université, CNRS 75252 Paris, France

Correspondence to: Yann Quilcaille (yann.quilcaille@env.ethz.ch)

Thank you again to the Editor for the recommendations. The manuscript has been modified as follows:

*1. I only realize now that neither the code nor the data is properly archived and therefore does not comply with the GMD code and data policy (https://www.geoscientific-model-development.net/policies/code_and_data_policy.html#item3). This can be remedied, e.g. by using Zenodo / Zenodo's github integration.*

In the section "Code and Data Availability", we have added the tag of the code on GitHub, as frozen versions for all users. We thus added the tag for the code of OSCAR and for the code used for all other operations.
We also added to the GitHub the parameters, masks and weights used.
Besides, we have replaced the URL for the availability of the data by a proper reference with its DOI.

*2. There are several instances of "Error! Reference source not found". Please fix.*
All of them have been fixed, thank you for pointing them out. Additional fixes on cross-references of figures have been performed.